# Oncogenic c-Myc induces replication stress by increasing cohesins chromatin occupancy in a CTCF-dependent manner

Silvia Peripolli[1], Leticia Meneguello[1,2], Chiara Perrod[1], Tanya Singh[1], Harshil Patel [3], Sazia T. Rahman [1], Koshiro Kiso[1], Peter Thorpe[4], Vincenzo Calvanese[1], Cosetta Bertoli [1,5] ✉ & Robertus A. M. de Bruin [1,2,5] ✉

Oncogene-induced replication stress is a crucial driver of genomic instability and one of the key events contributing to the onset and evolution of cancer. Despite its critical role in cancer, the mechanisms that generate oncogene-induced replication stress remain not fully understood. Here, we report that an oncogenic c-Myc-dependent increase in cohesins on DNA contributes to the induction of replication stress. Accumulation of cohesins on chromatin is not sufficient to cause replication stress, but also requires cohesins to accumulate at specific sites in a CTCF-dependent manner. We propose that the increased accumulation of cohesins at CTCF site interferes with the progression of replication forks, contributing to oncogene-induced replication stress. This is different from, and independent of, previously suggested mechanisms of oncogene-induced replication stress. This, together with the reported protective role of cohesins in preventing replication stress-induced DNA damage, supports a double-edge involvement of cohesins in causing and tolerating oncogene-induced replication stress.

DNA replication stress (RS) results from inefficient DNA replication, associated with slowing down and/or stalling of DNA replication forks, thus compromising the fidelity and timely completion of genome duplication[1]. RS is a common feature of cancer cells with accelerated S-phase entry, driven by oncogene activation, such as MYC, Ras and Cyclin E, or the absence of tumour suppressors, such as RB1[2]. Oncogene-induced RS directly contributes to the generation of genome instability[1] and Chromosomal instability (CIN)[3]. Importantly, recent work shows that CIN in turn generates RS[4]. This establishes a vicious cycle of RS-driven CIN where oncogene-induced RS contributes to the initiation of cancer and rapid genome evolution observed in tumours. Despite the critical role of RS in cancer, the mechanisms that generate oncogene-induced RS remains not fully understood (Fig. 1a). The most widely reported mechanisms are an increased occurrence of replication forks colliding with transcriptional bubbles, known as transcription–replication conflicts, and the dysregulation of replication initiation[1]. In the case of oncogenic overexpression of Cyclin E, both these mechanisms have been reported[5,6,7]. The reduced length of G1 following Cyclin E overexpression has been associated with a decrease in licensing events, which is thought to cause under-replication[6]. Cyclin E overexpression has also been shown to cause RS by increased transcription–replication conflicts in transcribed genes[5,7]. In contrast, the oncogenic activity of the transcription factor c-Myc has been reported to increase replication initiation events, thus causing over-replication[8,9]. Surprisingly, while c-Myc is thought to induce a large transcriptional programme to promote proliferation and growth[10], it has not been linked to increased transcription-replication interference.

Besides transcription machineries, other large protein complexes bound to DNA could interfere with replisome progression. The cohesin

[1]Laboratory Molecular Cell Biology, University College London, Gower Street, London, UK. [2]UCL Cancer Institute, University College London, Gower Street, London, UK. [3]Francis Crick Institute, Midland Rd 1, London, UK. [4]Queen Mary University, Mile End Road, London, UK. [5]These authors jointly supervised this work: Cosetta Bertoli, Robertus A. M. de Bruin. ✉e-mail: csttbrt@gmail.com; r.debruin@ucl.ac.uk

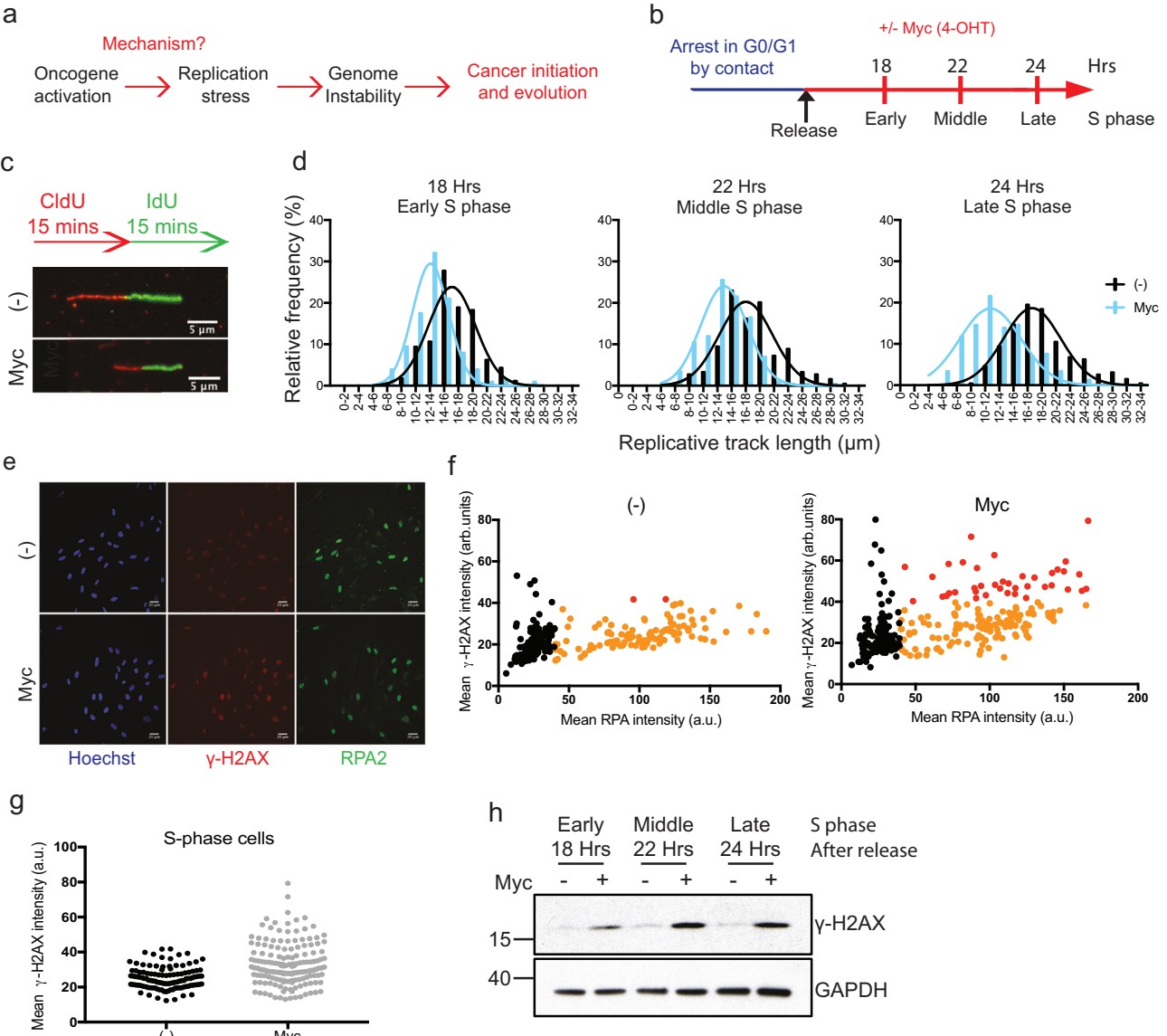

**Fig. 1 | c-Myc-induced replication stress and DNA damage depend on events in G1 phase. a** Schematic of oncogene-induced replication stress. **b** Schematic of the synchronisation experiments for G1-S release. RPE1 c-MycER cells were left to grow to confluence, then trypsinised and plated in fresh medium. After cell spreading, 4OH-T was added to induce c-Myc or left untreated as control. Samples were taken at indicated times after release representing early = 18 h, middle = 22 h, late = 24 h S-phase populations. **c** DNA fibre analysis of synchronised cells. Schematic showing the pulse labelling with the two nucleotide analogues. Immunofluorescence of representative fibres. **d** Histograms reporting the distribution of fibre length for control and c-Myc-induced cells at indicated times after release from arrest; *P* value****<0.0001 calculated with Mann–Whitney test. Representative of *n* = 2 experiments. **e** Representative images of RPA and γH2AX immunofluorescence. **f** Immunofluorescence staining of chromatin-bound RPA and γH2AX after 24 h release from confluence arrest. Scatter plot showing the intensity of RPA and γH2AX signal in single nuclei. Black = RPA-negative cells, orange = RPA-positive cells, red = RPA-positive cells with higher γH2AX signal. Representative of *n* = 2 experiments. **g** Graph showing γH2AX intensity in individual S-phase cells after 24 h release from confluence arrest plotted in the scatter plot. *P* value****<0.0001 calculated with Mann–Whitney test. Representative of *n* = 2 experiments. **h** Western blot of γH2AX at the indicated timepoints after release from G1 arrest, with and without c-Myc activation; early = 18 h, middle = 22 h, late = 24 h; GAPDH is a loading control. Representative of *n* = 3 experiments. Source data are provided as a Source data file.

complex is probably one of the most abundant protein complexes interacting with the DNA. Whilst previous work has established an important role for cohesins in the recovery from RS, preventing RS-induced DNA damage[11,12,13], more recent work indicates that cohesins could also slow down replisome progression during S-phase[14,15]. Cohesins are ring-shaped multiprotein complexes comprising two major subunits, Structural Maintenance of Chromosome (SMC) 1 and SMC3, along with the kleisin subunit Rad21 and STAG1 and STAG2 in mammalian cells[16]. The loading of cohesins onto DNA is highly regulated, and in mammalian cells depends on the activity of the loaders MAU2 and Nipped-B-like protein (NIPBL). The activity of the loaders is

antagonised by the release factor WAPL[17]. While loading and release occur throughout the entire cell cycle, during S-phase and G2 cohesins interaction with the DNA is more stable[18]. This is due to the establishment factors, ESCO1 and ESCO2 in mammalian cells, which acetylate the SMC3 subunit of the cohesin complex[19], and thus prevent the release activity of WAPL[20].

Recent evidence confirmed that in mammalian cells, as previously reported in budding yeast[21], cohesin rings are able to move along the DNA in a transcription-dependent manner[22]. Binding of the CCCTC-Binding Factor (CTCF) to CTCF sites is involved in the organisation of spatially interacting regions of chromatin[23], known as topologically

associated domains (TADs)[23,24]. It has been shown that CTCF sites can act as a road-block for cohesins, with accumulation of cohesins often detected at CTCF sites in mammalian cells[25,26]. While the majority of cohesins interact dynamically with the DNA, some are associated more stably, and reside at CTCF sites, where they participate in the organisation of chromatin loops[27].

Here, we report that c-Myc induces replication stress via increasing the number of cohesins bound to chromatin accumulating at sites in a CTCF-dependent manner, which represents a different mechanism for oncogene-induced RS to previously suggested mechanisms. Since MYC is hyper-activated in many cancers, this mechanism of oncogene-induced replication stress is likely to have an important role in cancer biology and potentially therapy.

## Results

Different oncogene-induced RS mechanisms have been proposed. To establish which of these contribute to MYC-induced RS, we exploited a c-Myc-ER inducible RPE1 hTERT cell line[28]. In this system, oncogenic c-Myc activation depends on the translocation of the c-Myc-ER protein into the nucleus after 4OH tamoxifen (4OH-T) addition (Supplementary Fig. 1a, b), which is independent of c-Myc-ER protein levels (Supplementary Fig. 1c). Twenty-four to 48 h after 4OH-T addition, oncogenic c-Myc activity can be observed via a decrease in colony formation (Supplementary Fig. 1d) and increased gene expression of well-known MYC target genes at both RNA (Supplementary Fig. 1e, f) and protein level (Supplementary Fig. 1g), and a progressive proliferative arrest of cells where c-Myc is induced (Supplementary Fig. 1h). As shown previously[28], activation of c-Myc-ER by 4OH-T induces RS (pCHK1 and pRPA) and DNA damage (pCHK2, p21 and γH2AX) response markers within 24 h (Supplementary Fig. 1i, j) and causes shortening of DNA fibre length (Supplementary Fig. 1k), indicative of slowing down of replication forks. Together these data support the use of this system to study the mechanisms underlying RS generation following acute oncogenic MYC induction.

### Oncogenic c-Myc activity in G1 phase is required to induce replication stress in S-phase

The proposed mechanism of oncogene-induced RS includes under-replication, which requires shorting of the G1 phase, over-replication and transcription–replication conflicts during S-phase. To gain some initial insights into how c-Myc induces RS we first established whether c-Myc activity in G1 and/or S-phase is required or sufficient to induce RS. We arrested cells in different cell cycle phases, released them with or without c-Myc activation and tested the levels of RS and DNA damage in the following S-phase. First, we used confluency to arrest cells in G0/G1. After release into the cell cycle, c-Myc was activated by adding 4OH-T or left untreated as control (Fig. 1b and Supplementary Fig. 1l), allowing us to study the first G1 and S phases (early, middle, and late S-phase, 18 hrs, 22 hrs and 24 hrs after release respectively) after oncogene activation. To measure RS, we analysed the length of DNA fibres, as a proxy for the speed of replication forks. The activation of c-Myc reduces the average DNA fibre length compared to control, suggestive of slowing down of replication forks (Fig. 1c, d and Supplementary Fig. 1m). We then measured DNA damage by monitoring the phosphorylation of the histone H2AX. We observed increased levels of the DNA damage marker γ-H2AX in S-phase cells by both Western blot and immunofluorescence (Fig. 1e–h). We confirmed these findings by synchronising the cells via nocodazole shake-off (Supplementary Fig. 1n–r). Together, these data suggest that the activation of c-Myc during G1 and S-phase can induce RS and DNA damage.

To test if G1 phase is required for c-Myc to induce RS, we synchronised cells in early S-phase by adding hydroxyurea (HU). Subsequently, we washed out HU to allow S-phase progression with or without c-Myc activation and analysed the levels of RS and DNA damage (Supplementary Fig. 2a, b). We did not observe any decrease in

DNA fibre length in c-Myc-activated cells compared to the control (Supplementary Fig. 2c). On the contrary DNA fibres appear slightly longer upon c-Myc induction, indicating that c-Myc activity during S-phase does not cause RS, but might even increase the replication capacity of the cell as reported in ref. 29. Correspondingly, we did not observe any increase in DNA damage in c-Myc cells (Supplementary Fig. 2d). As expected, c-Myc activation increased Cyclin E levels, a transcriptional target of MYC (Supplementary Fig. 2e). Since HU treatment by itself causes RS, which might mask c-Myc-induced RS, we also released cells synchronously into S-phase after a G1 arrest via CDK4/6 inhibition by Palbociclib, as reported by Trotter et al.[30] c-Myc was activated, via addition of 4OH-T, either immediately after release (18 and 21 h), therefore throughout G1 phase, or immediately before entering S-phase (4.5 and 7.5 h), and DNA damage and DNA fibres length were analysed as above (Supplementary Fig. 2g–j). As in the previous experiments, we observed DNA damage when cells experienced c-Myc activity during G1, confirming that c-Myc activity during G1 is required to generate RS in S-phase.

### Under-replication and replication–transcription collisions have a limited contribution to c-Myc-induced RS

A possible cause of RS could be a decrease in G1 length, which has been associated with reduced origin licensing and consequent RS upon Cyclin E overexpression[6]. We evaluated replication origin licensing in pre-extracted samples by measuring chromatin-bound Mcm7, a component of the helicase complex that is loaded on the DNA during G1 as in ref. 6. We did not observe a decrease in origin licensing (Supplementary Fig. 2k), indicating that reduced origin licensing is an unlikely causative mechanism for c-Myc-induced RS.

It has been previously reported that RS can result from increased transcription–replication collisions[5,7]. To test if interference between replication and c-Myc-induced transcription contributes to RS we decreased global transcription levels via treatment with the RNA polymerase inhibitor 5,6-Dichloro-1-β-d-Ribofuranosyl Benzimidazole (DRB) for 2 h, as in ref. 31, during the first S-phase after release from G1 (Supplementary Fig. 2l, m). We confirmed that c-Myc activation increases global transcription levels, by 5-Ethynyl Uridine (EU) incorporation, and that DRB reduces it (Supplementary Fig. 2l). However, DNA fibre analysis shows that this does not rescue the c-Myc-dependent decrease in DNA fibre length (Supplementary Fig. 2m). This suggests that in this system transcription inhibition by DRB does not prevent c-Myc-induced RS in the first S-phase, though we cannot exclude a role for replication–transcription collisions in other settings.

### c-Myc activation increases cohesins on chromatin

We next focused on alternative mechanisms that could contribute to c-Myc-induced RS. A potential source of RS could be protein complexes interacting with the DNA during S-phase that could slow down replication fork progression. We hypothesised that potential candidates for this could be the cohesin complexes. Whilst cohesin has been shown to have a role in preventing RS-induced DNA damage[11], we hypothesised that a c-Myc-induced increase in cohesin chromatin occupancy during G1 could slow down replisome progression during S-phase as suggested in refs. 14 and 15. We therefore analysed the fraction of chromatin-bound cohesin subunits SMC1 and SMC3 by Immunofluorescence (IF) of pre-extracted samples, in cells released from G0/G1 with and without c-Myc. These data show that higher levels of SMC1 and SMC3 are detected on chromatin in c-Myc-activated cells both in G1 and in S-phase (Fig. 2a and Supplementary Fig. 3a). This was confirmed in asynchronous cell populations (Supplementary Fig. 3b) and by chromatin preparations followed by Western blot analysis (Fig. 2b).

Our IF analysis on pre-extracted nuclei together with the chromatin preparation and WB analysis establish an overall increase in chromatin occupancy of the cohesin subunits SMC1 and SMC3, but do

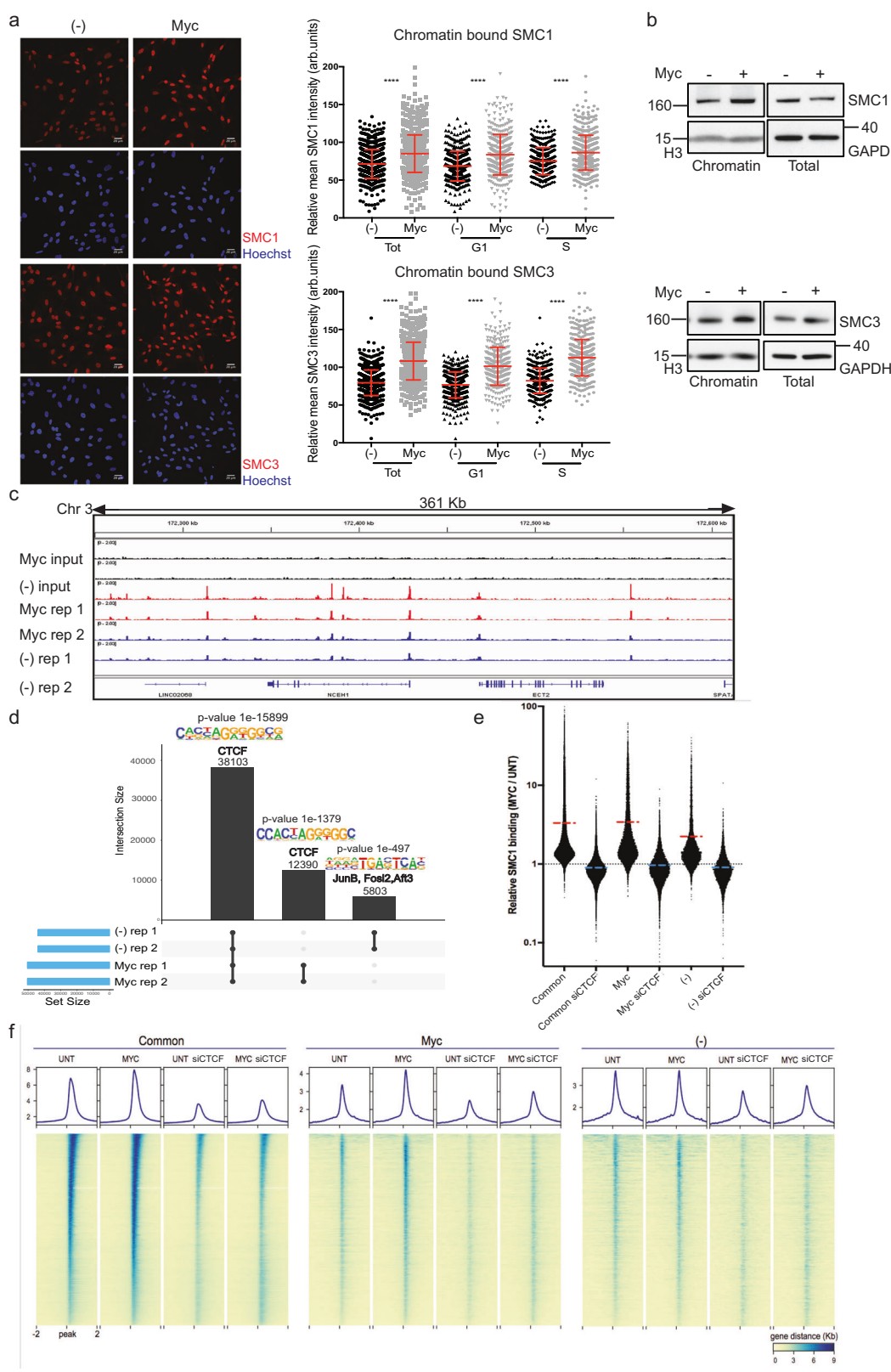

not reveal where these accumulate in the genome. To investigate if this c-Myc-dependent increase in chromatin-bound cohesin accumulates at specific sites, we established genome-wide binding of the cohesin component SMC1 by ChIP-seq in cells with activated c-Myc or control (Fig. 2c–e). We performed these experiments at 48 h after c-Myc activation, when we observe consistent levels of RS (Supplementary Fig. 1k). These experiments show that cohesin accumulates at 50,493

sites in c-Myc cells compared to 43,906 in control (Fig. 2d, see set size). Most peaks (38,103) are common to control and c-Myc-activated cells, with 12,390 unique to c-Myc and 5803 only present in control (Fig. 2d). Motif analysis of the common peaks shows enrichment of CTCF consensus, which is in line with previously reported cohesin accumulation at CTCF sites[25]. The c-Myc-specific peaks, not detected in control samples, are also enriched for CTCF motif. This suggests that the

**Fig. 2 | c-Myc activation increases cohesion chromatin occupancy in G1.**
**a** Synchronised cells were released into the cell cycle and immunofluorescence of chromatin-bound cohesin subunits SMC1 and SMC3 were performed at 18 h after release. Left panels; representative images of SMC1, SMC3 and Hoechst. Right: graph reporting the intensity of SMC1 and SMC3 signals in total, S-phase and G1-phase single nuclei of untreated and c-Myc cells. *P* value****<0.0001 calculated with Mann–Whitney test. Pool of *n* = 3 experiments. **b** Western blot of chromatin pre-parations and total cell lysates in asynchronous population with and without c-Myc activation for 16 h. GAPDH and H3 are loading controls. Representative of *n* = 3 experiments. **c** Binding of SMC1 to the reported DNA loci in untreated and c-Myc-

activated cells. Two repeats for each condition are represented. **d** Graph repre-senting the analysis of SMC1 binding distribution in c-Myc and untreated cells. Binding motif prediction with *P* value for each group, along with the published similar consensus identified. **e** Quantification of SMC1 binding in common, c-MYC-only or untreated (−) only peaks in c-Myc-activated cells relative to the untreated cells. Dash line represents the mean (red for sicontrol; blue for siCTCF). **f** Average profile and heatmap of SMC1-bound regions (ChIP-seq) in the groups identified in (**d**): common peaks, c-Myc only and untreated only (−) in untreated (UNT) of c-Myc-activated (C-MYC) cells, with sicontrol or CTCF-downregulation (siCTCF). Source data are provided as a Source data file.

increased chromatin-bound cohesins, detected in chromatin preps from c-Myc cells, are likely to accumulate at CTCF sites. To further evaluate this, we carried out quantitative ChIP-seq of SMC1 pulldowns spiked with Drosophila chromatin in each sample, as described in ref. 32, to allow for signal normalisation across all samples. Quantitative SMC1 binding was performed in c-Myc and control cells in both siCTCF and sicontrol treated cells to establish CTCF dependence. These data show that relative SMC1 binding (c-Myc/control) at all sites is higher in c-Myc cells compared to control cells and that this is CTCF-dependent (Fig. 2e, f). These data indicate that c-Myc increases cohe-sins at pre-existing sites, which, as previously shown, is predominantly CTCF-dependent. The increased cohesin binding at CTCF sites upon c-Myc activation was confirmed by ChIP quantitative PCR (Supple-mentary Fig. 3c).

Together, these data suggest that the c-Myc-induced accumula-tion of cohesins on chromatin takes place mainly at pre-existing CTCF sites.

**Preventing cohesin accumulation on chromatin reduces c-Myc-induced replication stress**
Next, we wanted to investigate if the increased presence of cohesins on chromatin is required for c-Myc-induced RS. We tested this by redu-cing the levels of cohesins on DNA by reducing the levels of the cohesin component Rad21. We used a non-efficient siRNA to ensure that there is a reduction of Rad21, rather than a complete loss, to prevent cell cycle defects during the first S-phase (Supplementary Fig. 3d). Analysis of SMC1 binding to chromatin confirms that Rad21 knockdown redu-ces the levels of chromatin-bound cohesins both in untreated and in c-Myc cells (Fig. 3a, b). Importantly, the levels of SMC1 on chromatin in Rad21-depleted c-Myc cells are similar to the untreated control, allowing us to test if the increased chromatin binding of cohesins is at the basis of c-Myc-induced RS. Reducing the levels of cohesins on chromatin in cells experiencing oncogenic c-Myc increases DNA fibre length in the first S-phase in both synchronous and asynchronous cell populations (Fig. 3c, d and Supplementary Fig. 3e, f), indicative of a reduced level of RS. Rad21 depletion did not affect the expression level of well-established MYC target genes (Supplementary Fig. 3i). These data support the idea that the increase of cohesins on DNA is required for c-Myc-induced RS.

**Accumulation of cohesins at CTCF sites is required for c-Myc-induced replication stress**
Next, we tested if a c-Myc-dependent increase in chromatin-bound cohesins is sufficient to cause replication stress or whether it requires accumulation at CTCF sites. Strikingly, depleting CTCF (Supplemen-tary Fig. 3j) completely rescued RS induced by c-Myc, without reducing the global amount of cohesins on DNA or affecting the cell cycle profiles (Fig. 3e–h and Supplementary Fig. 3k, l). These data indicate that, whilst an increase in chromatin-bound cohesins is required, c-Myc-induced RS depends on CTCF. We analysed the RS and DNA damage response activation in these cells, following Rad21 and CTCF depletion. c-Myc activation increases DNA damage signalling, upon

Rad21 and CTCF depletion we observed a significant decrease in both CHK1 phosphorylation and γH2AX levels compared to control silen-cing, suggesting that in both cases RS-induced DNA damage was reduced (Supplementary Fig. 3m, n). Together these data indicate that c-Myc leads to an increased accumulating of cohesins at specific sites in a CTCF-dependent manner, thus causing a slowdown of replication forks to generate RS.

**c-Myc activation increases the cohesin loader MAU2 protein levels**
To investigate how c-Myc could increase cohesin chromatin occu-pancy, we tested if its activation affects the expression levels of cohesin subunits and regulators. SMC1, SMC3 and Rad21 protein levels did not change significantly upon c-Myc activation (Supplementary Fig. 4a). Interestingly, protein levels of the cohesin loader MAU2 increased upon c-Myc activation in both synchronised (Fig. 4a) and asynchronous cells (Fig. 4b and Supplementary Fig. 4b), which corre-sponds to an increase in mRNA levels (Supplementary Fig. 4c, d). In vertebrates, cohesin loading requires NIPBL and MAU2[33,34]. While NIPBL is the proper cohesin loader, MAU2 stabilises the protein levels of NIPBL[35,25,36], therefore we analysed NIPBL levels in c-Myc-activated cells. Both MAU2 and NIPBL protein levels increase upon c-Myc acti-vation (Supplementary Fig. 4b), while NIPBL mRNA does not change significantly (Supplementary Fig. 4d), suggesting that a c-Myc-dependent increase in MAU2 expression could stabilise NIPBL pro-tein levels.

To establish if a c-Myc-induced increase in cohesin chromatin occupancy could depend on the upregulation of MAU2, we reduced MAU2 levels in c-Myc-activated cells to control levels using a non-efficient siRNA (Fig. 4c) and tested the presence of cohesins on chro-matin and RS levels. Like for Rad21, reducing MAU2 accumulation prevents excess cohesin loading onto chromatin (Fig. 4d, e) and RS upon c-Myc activation in both synchronised (Fig. 4f, g and Supple-mentary Fig. 4e) and asynchronous cells (Supplementary Figs. 4f and 3l, m). As for Rad21 knockdown experiments, cell cycle distribution is not affected by MAU2 depletion (Supplementary Fig. 4g). These data indicate that c-Myc-dependent increase in MAU2 levels could be at the basis of the increased loading of cohesins on chromatin, which causes RS during S-phase.

To investigate if a c-Myc-dependent increase in MAU2 levels could contribute to the generation of RS, we transiently expressed MAU2-GFP, or GFP as control, in RPE1 cells and analysed RPA phosphoryla-tion, marker of RS, and γH2AX, marker of DNA damage, by quantitative immunofluorescence and Western blot (Supplementary Fig. 4h–m). Whilst both markers increase in IF in MAU2 transfected cells, an increase in phosphorylation of RPA is particularly pronounced in IF and total lysates. These data suggest that increased expression of MAU2 can cause some RS and RS-induced DNA damage. While this provides an initial indication of a potential mechanism by which c-Myc increases the chromatin occupancy of cohesins, this data is not exhaustive and needs further investigation, which will be the focus of future work.

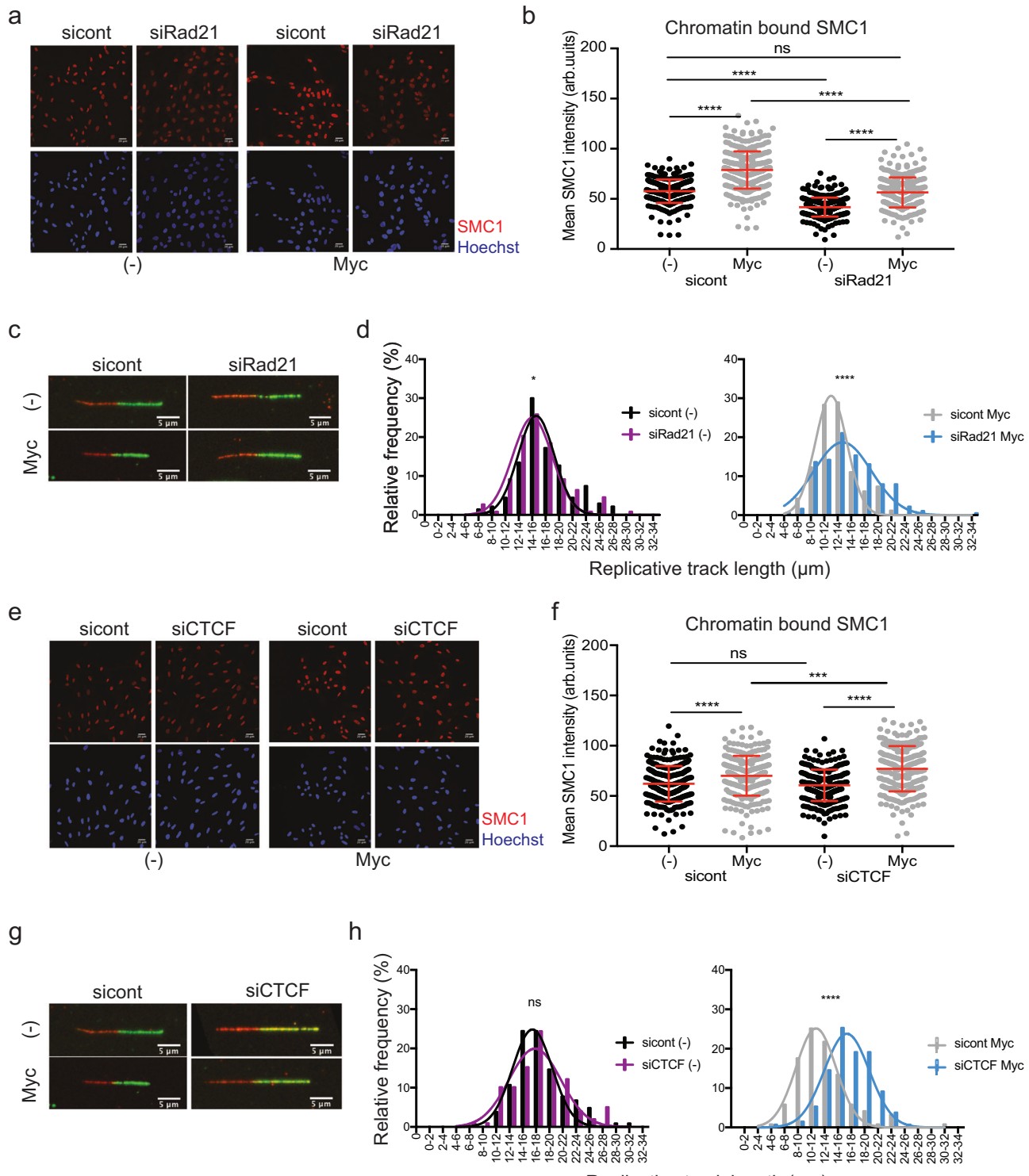

**Fig. 3 | Reducing the levels of cohesin on chromatin at CTCF sites prevents c-Myc-induced replication stress. a** Immunofluorescence showing chromatin-bound SMC1 in synchronised sicontrol and siRad21 depleted cells at 20 h after release, with and without c-Myc activation. **b** Graphs representing the intensity of SMC1 signal in immunofluorescence. *P* value ****<0.0001 calculated with Mann–Whitney test. Representative of *n* = 3 experiments. **c** Immunofluorescence of representative fibres in synchronised sicontrol and siRad21 depleted cells at 20 h after release from G1. **d** Histograms reporting the distribution of fibre length in synchronised sicontrol and siRad21 depleted cells. *P* value ****<0.0001 calculated with Mann–Whitney test. Representative of *n* = 3 experiments.

**e** Immunofluorescence showing chromatin-bound SMC1 in synchronised sicontrol and siCTCF-depleted cells at 20 h after release, with and without c-Myc activation. **f** Graphs representing the intensity of SMC1 signal in the immunofluorescence. *P* value ****<0.0001 calculated with Mann–Whitney test. Representative of *n* = 3 experiments. **g** Immunofluorescence of representative fibres in synchronised sicontrol and siCTCF-depleted cells at 20 h after release from G1. **h** Histograms reporting the distribution of fibre length in synchronised sicontrol and siCTCF-depleted cells. *P* value ****<0.0001 calculated with Mann–Whitney test. Representative of *n* = 3 experiments. Source data are provided as a Source data file.

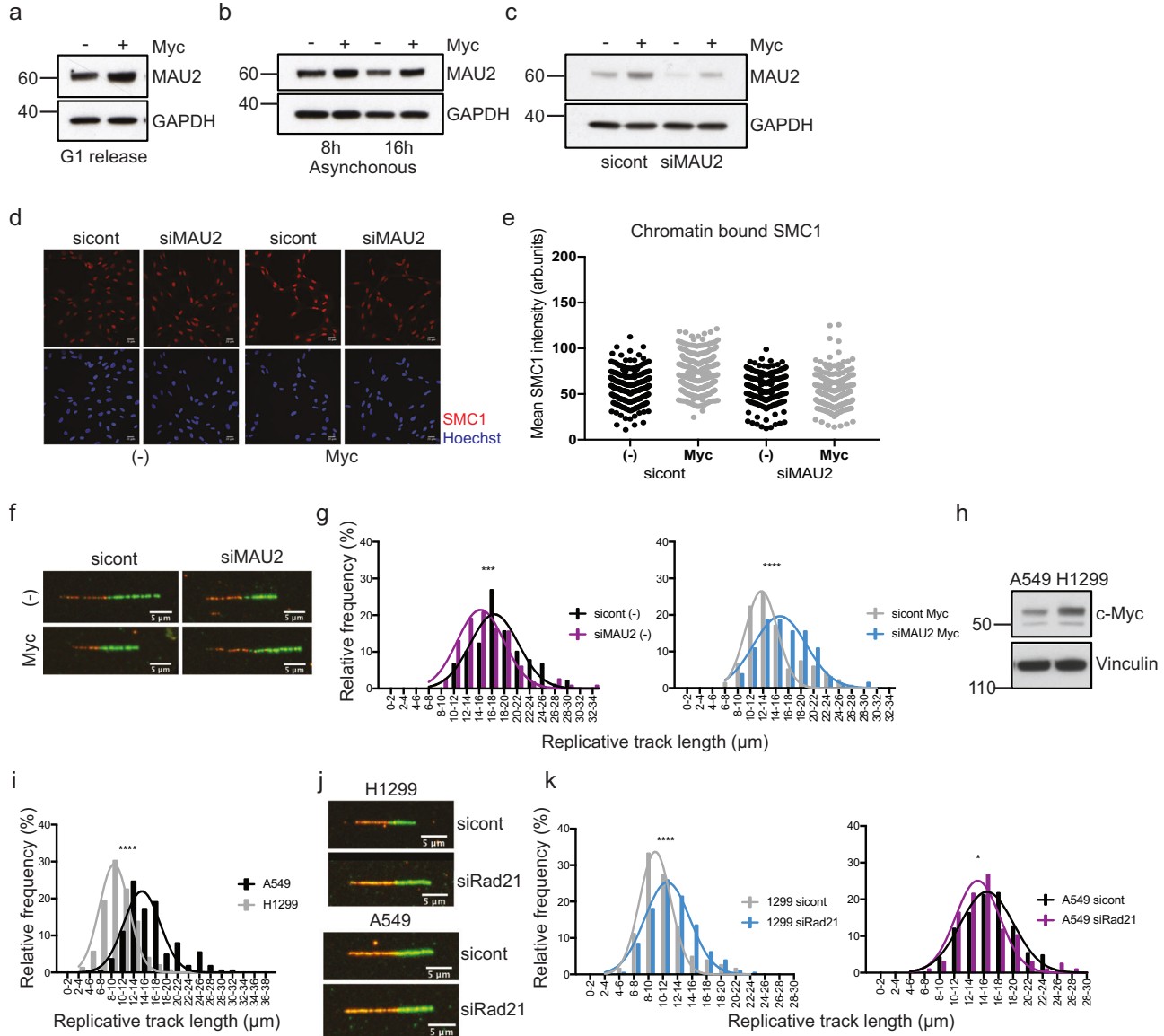

**Fig. 4 | Reducing the levels of cohesin loader MAU2 prevents c-Myc-induced replication stress. a** Western blot showing total levels of the cohesin loader Mau2 in synchronised cells at 20 h after release, with and without c-Myc activation. Representative of n = 3 experiments. GAPDH is a loading control. **b** Western blot showing total levels of the cohesin loader MAU2 in asynchronous population with and without c-Myc activation for the indicated timepoints. Representative of n = 2 experiments. GAPDH is a loading control. **c** Western blot showing MAU2 knock-down in synchronised cells at 20 h after release, with and without c-Myc activation. Representative of n = 3 experiments. GAPDH is a loading control.
**d** Immunofluorescence showing chromatin-bound SMC1 in synchronised sicontrol and siMAU2-depleted cells at 20 h after release, with and without c-Myc activation. **e** Graphs representing the intensity of SMC1 signal in the immunofluorescence. *P* value ****<0.0001, ***=0.0001 calculated with Mann–Whitney test. Representative

of n = 2 experiments. **f** Immunofluorescence of representative fibres in synchronised sicontrol and siMau2-depleted cells at 20 h after release from G1.
**g** Histograms reporting the distribution of fibre length in synchronised sicontrol and siMau2-depleted cells. *P* value ****<0.0001 calculated with Mann–Whitney test. Representative of n = 3 experiments. **h** Western blot showing total levels of c-Myc in A549 and H1299 cells. Representative of n = 3 experiments. GAPDH is a loading control. **i** Histograms reporting the distribution of fibre length in A549 and H1299 cells. *P* value ****<0.0001 calculated with Mann–Whitney test. Representative of n = 3 experiments. **j** Immunofluorescence of representative fibres in A549 and H1299 cells after Rad21 depletion. **k** Histograms reporting the distribution of fibre length in A549 and H1299 cells after Rad21 depletion. *P* value*=0.0356, ****<0.0001 calculated with Mann–Whitney test. Representative of n = 3 experiments. Source data are provided as a Source data file.

## Depletion of the cohesins subunit Rad21 decreases replication stress in cancer cells overexpressing c-Myc

We finally tested whether this mechanism of c-Myc-induced RS could potentially contribute to RS in cancer cells (Fig. 4h–k). We selected lung cancer cell lines expressing different levels of c-Myc and measured the length of DNA fibres to evaluate the presence of RS. We observed reduced fibre length in the H1299 cell line compared to the A549 cells, which correlates with higher levels of c-Myc in H1299 cells (Fig. 4h, i). To establish if a reduction in cohesins can

rescue RS levels we depleted Rad21 (Supplementary Fig. 4n) and measured DNA fibre length in both cell lines (Fig. 4j, k and Supplementary Fig. 4o). Rad21 depletion increased fibre length only in the H1299 cells, where c-Myc is highly expressed, but not in A549 cells. This supports a causative role for cohesins in casing RS in cells overexpressing c-Myc, which agrees with our findings. Interestingly, depletion of Rad21 in the A549 cells, which do not experience c-Myc-induced RS, partially reduces fibre length. This is in line with a protective role for cohesins in preventing RS, which agrees with

reported work, supporting a double-edge involvement of cohesins in causing and tolerating RS.

## Discussion

Here, we investigate the mechanisms by which oncogenic MYC induces replication stress. Our data show that a c-Myc-induced increase in cohesins on the DNA contributes to the induction of RS. This is different from previously reported mechanisms of oncogene-induced RS, which are linked to deregulation of replication origin usage and/or transcription-replication conflicts. Firstly, we show that activation of c-Myc increases chromatin-bound cohesins levels before S-phase entry and that restoring the amount of cohesins bound to chromatin to control levels in cells experiencing oncogenic c-Myc prevents RS. Secondly, we show that c-Myc-dependent accumulation of cohesins on chromatin is not sufficient to cause RS, but also requires CTCF to facilitate the accumulation of cohesins at CTCF sites. Thirdly, the c-Myc-dependent induction of the cohesion loader MAU2 provides a potential mechanism through which c-Myc affects cohesin regulation. Finally, we show a causative role for cohesins in causing RS in cancer cells overexpressing c-Myc. Together our data support an important role for cohesins in causing oncogene-induced RS, in addition to their role in RS-induced DNA damage repair. Since MYC activation is a crucial event in many human cancers[37], identifying the mechanisms through which this oncogene promotes RS provides critical insights into cancer biology.

Our findings are surprising in light of previous work that indicates an important role for cohesins in preventing RS and DNA damage[12,13]. Based on our data, we speculate that whilst the presence of chromatin-bound cohesins during S-phase is required to protect stalled forks and repair damaged DNA, in an oncogenic context excessive cohesin loading and its subsequent hyperaccumulation at CTCF-dependent sites can interfere with the progression of the replisome. This is in agreement with recently published work in mammalian cells which shows that increased presence of cohesins on DNA slows down fork progression[15] and work in yeast, which shows that DNA damage accumulates in SMC-rich genomic regions during replication[38].

Given that oncogene-induced replication stress is a crucial driver of genomic instability and one of the key events contributing to the onset of cancer, our work provides additional mechanistic insight into cancer biology and potentially therapy.

## Methods

### Cell culture and treatments

Cell lines used were immortalised hTERT human Retinal Pigment Epithelia 1 (ATCC CRL-4000) c-Myc-ER cells (previously described in ref. 1) and Retinal Pigment Epithelia 1 (ATCC CRL-4000) ER empty. Cells were cultured in phenol red-free DMEM/F12 media supplemented with 10% charcoal-treated foetal bovine serum, 1% penicillin/streptomycin (Gibco) and 3% sodium bicarbonate (Gibco). Cells were maintained in Puromycin (2 µg/ml). Cells were treated with 4-hydroxytamoxyfen (4OH-T) (100 nM), with HU overnight (2 mM) and for 4 h (0.2 mM), with Nocodazole for 8 h (200 ng/ml), with Palbociclib for 24 h (150 nM) and with DRB for 2 h (75 µM).

RPE1 hTERT (ATCC CRL-4000) cells were cultured in DMEM/F12 media supplemented with 10% foetal bovine serum, 1% penicillin/streptomycin (Gibco) and 3% sodium bicarbonate (Gibco).

A549 (ATCC CCL-185) cells were cultured with DMEM/F12 media supplemented with 10 % FBS and 1% penicillin/streptomycin (Gibco). H1299 (ATCC CRL-5803) cells were cultured in RPMI media supplemented with 10% FBS and 1% penicillin/streptomycin (Gibco).

### siRNA transfection

For siRNA transfection, Lipofectamine RNAiMAX (Invitrogen, 13778-075) was used following the manufacturer's instructions. Experiments were carried out 20 h after retro-transfection for synchronised cells. Unsynchronised cells were split 24 h after transfection and then used for experiments 24 h later.

siRNA oligonucleotides with the following sequences were used: siRad21 (GACCAAGGUUCCAUAUUAU), siCTCF (GGAGCCUGCCUGC CGUAGAAAUUTT), siMAU2 (CCUCAGAACUUAACAUCUG). Non-targeting siRNA, referred as to sicont, (D-001206-13-05 siGENOME Non-Targeting siRNA pool) was purchased from Dharmacon.

### Plasmid transfection

For transient plasmid transfection, Lipofectamine 2000 (Invitrogen, 11668019) was used following the manufacturer's instructions. The plasmids used were pEGFP N1 and Scc4-Flag-EGFP[2].

### Immunofluorescence

When appropriate cells were pre-extracted with ice-cold 0.2% triton solution in PBS 1× for 1 min and fixed for 20 min in formaldehyde solution 4%. If the extraction protocol was not carried out, cells were permeabilised for 4 min in 0.2% triton solution in PBS 1× after fixation. Cells were blocked in blocking solution (1% BSA, 0.2% Tween in PBS) for 1 h at room temperature. The incubation with the primary antibodies anti-RPA2 (Millipore RPA34-20 1:500), anti-Phospho-Histone H2A.X (γH2AX) (Ser139) (Cell Signaling Technology g-H2AX 20E3 1:250), anti-SMC1 (Bethyl laboratories A300-055A 1:1000), anti-SMC3 (Bethyl laboratories A300-060A 1:2000), anti-Mcm7 (Santa Cruz Biotechnology sc-56324 1:150), anti-RPA32 Phospho S4/S8 (Bethyl laboratories A300-245A 1:500) and anti-GFP (Abcam AB1218 1:1000) was carried out overnight. The following day the coverslips were incubated with secondary antibodies (Alexa Fluor® 488 goat anti-mouse 1:2000 and Alexa Fluor® 647 goat anti-rabbit 1:2000) for 2 h at room temperature. The cells were stained with Hoechst (Invitrogen) solution 1:10,000 in PBS 1×. The coverslips were mounted on slides with mounting medium Fluoroshied (Sigma). Images were obtained with Leica SPE2 40x objective lens and processed with Fiji.

For the quantitative analysis, between 200 and 300 cells were analysed per sample.

### EdU incorporation

Cells on coverslips were incubated with EdU (final concentration 10 µM) for 30 min and fixed in 4% formaldehyde solution. Cells were permeabilised in 0.2% triton for 5 min and incubated with Click-it reaction cocktail (Click-it Alexa Fluor 647 C-10424 Invitrogen) for 30 min. Nuclei were stained with Hoechst (Invitrogen) solution 1:10,000. The coverslips were mounted on slides with mounting medium Fluoroshied (Sigma). Images were obtained with Leica SPE2 ×40 objective lens and processed with Fiji.

### EU incorporation

For detection of global RNA synthesis levels by 5-Ethynyl Uridine (EU) staining, 1 mm EU was added to cells for 1 h prior to collection. Cells were fixed in 4% formaldehyde. EU detection was performed using the Click-iT RNA Alexa Fluor 488 Imaging kit (ThermoFisher, C10329) following the manufacturer's instructions. Coverslips were rinsed for 2 min in Click-iT reaction rinse buffer and stained in Hoechst solution (1:10,000, Invitrogen H3570) for 5 min at room temperature. Fluoroshield (Sigma, F6182) was used for mounting on slides. Once dry, coverslips were sealed with nail varnish. Leica SPE2 using 40x objective lens and processed with Fiji.

### Fibre analysis

Cells were labelled with 25 µM CIdU for 15 min at 37 °C and then with 250 µM CO2-equilibrated IdU (final concentration 250 µM) for 15 min at 37 °C. Fibre spreading and labelling were performed as in[3]. The fibres were stained with primary antibodies (Rat anti-BrdU Abcam ab6326 1:250, Mouse anti-BrdU BD Biosciences 347580 1:100) overnight and with secondary antibodies (Alexafluor 555 goat anti-rat 1:500,

Alexafluor 488 goat anti-mouse1:500) for 1.5 h. Images were obtained with Leica SPE2 ×63 objective lens and processed with Fiji. 100–200 fibres were measured for each experiment.

Composite images were constructed to visualise red and green channels simultaneously. The 'line' tool in Fiji was used to measure length of DNA replicating fibres, which are characterised by the presence of consecutive red and green signals. The total amount of fibres (ongoing fibres, replication origins, replication terminations, stalled forks) and replication origins (characterised by a red track between two green tracks) were counted to quantify the percentage of origin firing.

## Chromatin preparation

RPE1 c-Myc ER cells were seeded in 10-cm dishes, and c-Myc expression was activated upon tamoxifen treatment. Cells were harvested after 16 h of c-Myc activation, and the chromatin was isolated with Chromatin Extraction Kit (ab117152, Abcam) according to the manufacturer's protocol. The sonication was performed in a Diagenode Bioruptor® sonicator using the programme 10 min: 30 s on, 30 s off.

## Western blot

Cell extracts were prepared in RIPA buffer (Tris pH 7.5 20 mM, NaCl 150 mM, EDTA 1 mM, EGTA 1 mM, NP-40 1%, NaDoc 1%), phosphatase inhibitor cocktail 2 and 3 (Sigma P5726 and P0044) 1:1000, and protease inhibitor cocktail (Sigma P8340) 1:1000. Primary antibodies used anti-Phospho-Histone H2A.X ($\gamma$H2AX) (Ser139) (Cell Signaling Technology g-H2AX 20E3 rabbit 1:250), anti-Scc4 (MAU2) (Abcam ab183033 1:500), anti-Idn3 (NIPBL) (Abcam ab106768 1:500), anti-SMC1 (Bethyl laboratories A300-055A rabbit 1:10,000), anti-SMC3 (Bethyl laboratories A300-060A rabbit 1:10,000), anti-Rad21 (Abcam ab992 1:2000), anti-Cyclin E (Santa Cruz Biotechnology HE12 sc-247 1:1000), anti-CTCF (Chip-grade AB70303 Abcam 1:5000), anti-RPA32 Phospho S4/S8 (Bethyl laboratories A300-245A 1:1000), anti-RPA2 (Millipore RPA34-20 1:1000), anti-c-Myc (Santa Cruz Biotechnology 9E10 sc-40 1:1000), anti-CHK1 (Cell Signalling Technology 2360 1:1000), anti-CHK1 Phospho S345 (Cell Signalling Technology 2341 1:250), anti-CHK2 (Cell Signalling Technology 2662 1:1000), anti-CHK2 Phospho Thr68 (Cell Signalling Technology 2661 1:1000), anti-p21 (Cell Signalling Technology 2947 1:1000), anti-MDM2 (Cell Signalling Technology 86934 1:1000). Secondary antibodies used were goat anti-mouse IgG HRP conjugate (Thermofisher Scientific PA1-74421 1:4000) and goat anti-rabbit IgG HRP conjugate (ThermoFisher Scientific 31460 1:4000). GAPDH (Genetex GT239 1:1000), Vinculin (Abcam AB129002 1:10000) and H3 (Cell Signalling Technology 9715 1:5000) were used as loading controls.

## Flow cytometry

For analysis of DNA content by propidium iodide (PI) staining, cells were collected by trypsinisation and fixed in 70% ethanol at −20 °C overnight. After centrifugation, the cell pellet was washed with PBS and resuspended in 100 mg/ml RNaseA and 50 mg/ml propidium iodide in PBS and incubated at 4 °C overnight.

For analysis using EdU/DAPI, cells were treated with EdU at the final concentration of 10 mM and incubated for 30 min at 37 °C with 5% $CO_2$. Cells were washed with cold PBS, collected with trypsin, and fixed in 4% formaldehyde for 15 min at room temperature. Next, cells were washed in 1 mg/ml BSA in PBS and resuspended in saponin-based permeabilization for 15 min followed by EdU detection using Click-iT EdU Alexa Fluor 647 Flow Cytometry Assay Kit (Invitrogen, C-10424) following the manufacturer's instructions. Cells were incubated in 0.5 mg/ml DAPI for at least 15 min at room temperature and then analysed.

For all flow cytometry analyses, samples were measured on a BD LSRII flow cytometer using DIVA software (BD) and analysed using FlowJo software.

## Survival assay

Cells were treated for 48 h with 4OH-T or left untreated. The same volume of cell suspension was re-plated in 5-cm dishes, and colonies were left to grow for one week. Cells were finally fixed and stained in 70% EtOH and 0.5% Methylene blue.

## ChIP-seq

Cells were cultured in 15-cm dishes for 48 h with or without the addition of 4OH-T. Cells were then washed with 1× PBS and crosslinked in 10 ml of 1% formaldehyde at RT for 10 min. Quenching was carried out by adding 1 ml of 1.25 M glycine for 10 min at RT. Cells were then scraped in PBS, spin down, resuspended in cold buffer A (100 mM Hepes pH8, 100 mM EDTA, 5 mM EGTA, 2.5% Triton) and rocked for 10 min at 4 °C. The same step was repeated using cold buffer B (100 mM Hepes pH 8, 2 M NaCl, 100 mM EDTA, 5 mM EGTA, 0.1% Triton). Cells were resuspended in cold ChIP buffer (25 mM tris/HCl pH8, 2 mM EDTA, 150 mM NaCl, 1% Triton, 0.1% SDS) plus protease inhibitor cock- tail (Sigma P8340) and sonicated at maximum output on a Bioruptor for 30 s on/30 s off for 30 min using Diagenode tubes. Sonication was checked on 1% agarose gel. After sonication, lysates where centrifuged for 15 min at maximum speed at 4 °C. Protein A solution was prepared by resuspending beads in ChIP buffer (about 50%) 1 $\mu$g/$\mu$l BSA and rocking at 4 °C for 15 min. The supernatant (soluble chromatin) was transferred in new tubes and pre-cleared adding blocked protein A solution and rocking for 2 h at 4 °C. Cleared soluble chromatin was centrifuged for 4 min at 4000 rpm at 4 °C. The supernatant was transferred in a new tube and quantified by Qubit (ThermoFisher) and 1% was saved as input.

The soluble chromatin was incubated overnight with 10 $\mu$g of anti-SMC1 (Bethyl laboratories A300-055A rabbit); for quantitative ChIP-seq (qChIP-seq), equal amount of chromatin (25 $\mu$g) were mixed with 20 ng of Spike-in Chromatin (from S2-Drosophila cell line, Active Motif #53083) and incubated overnight with 10 $\mu$g of anti-SMC1 together with 2 $\mu$g Spike-in Antibody (Active Motif #61686) The following day 20 $\mu$l protein A beads prepared as above, were added to chromatin, which was then rocked at 4 °C for 2 h. Beads were spin down for 2 min at 2000 rpm and washed with ChIP buffer, wash solution 1 (25 mM Tris/HCl pH8, 2 mM EDTA, 500 mM NaCl, 1% triton, 0.1% SDS), Wash solution 2 (250 mM LiCl, 1% NP-40, 1% NaDOC, 1 mM EDTA, 10 mM Tris/HCl pH 8) and twice with TE. TE was then removed and elution buffer (1% SDS, 100 mM NaHCO$_3$) was added. All samples were incubated at 65 °C overnight to reverse cross-linking. The day after, samples were purified using QIAquick PCR purification kit (QIAGEN). The DNA was then diluted in ddH$_2$O. QC was performed using Qubit (ThermoFisher) and either TapeStation (Agilent) or BioAnalyzer (Bio-Rad). The DNA samples were used to generate libraries using the KAPA HyperPrep kit (Roche) or the Ovation UltraLow V2 DNA-seq kit (Tecan) according to the manufacturer's instructions. The Illumina-compatible libraries were pooled to 4 nM, and sequencing with the HiSeq 4000 with at least 75 bp reads.

## ChIP-seq data analysis

The nf-core/chipseq pipeline (version 1.0.0, https://www.ncbi.nlm.nih.gov/pubmed/32055031; https://doi.org/10.5281/zenodo.3240507) written in the Nextflow domain-specific language (version 0.32.0[4]) was used to perform the primary analysis of the samples in conjunction with Singularity (version 2.6.0[5]). The command used was "nextflow run nf-core/chipseq --design design.csv --genome hg19 --singleEnd --narrowPeak --min_reps_consensus 2 -profile crick -r 1.0.0". To summarise, the pipeline performs adaptor trimming (Trim Galore!– https://www.bioinformatics.babraham.ac.uk/projects/trim_galore/), read alignment (BWA[6]) and filtering (SAMtools[7]; BEDTools[8]; BamTools[9]; picard-tools– https://github.com/pysam-developers/pysam; http://broadinstitute.github.io/picard), normalised coverage track generation (BEDTools[8]; bedGraphToBigWig[10]), peak calling (MACS[11]) and annotation relative to

gene features (HOMER[12]), consensus peak set creation (BEDTools[8]), differential binding analysis (featureCounts[13]; R[14]; DESeq2[15]) and extensive QC and version reporting (MultiQC[16]; FastQC–https://www.bioinformatics.babraham.ac.uk/projects/fastqc/; preseq[17]; deepTools[18]; phantompeakqualtools[19]). All data was processed relative to the human UCSC hg19 genome (UCSC[20]) downloaded from AWS-iGenomes (https://github.com/ewels/AWS-iGenomes). Gene annotation files in GTF format were originally downloaded from UCSC on July 17, 2015.

Peaks and bigWig coverage files were manually inspected in the Integrative Genomics Viewer (IGV) genome browser[21]. A selection of these regions (Fig. 3a) showing peaks in the c-Myc-activated samples compared to the control. Three peak sets were generated based on being common to both or exclusively present in either one of the c-Myc activated or control conditions (Fig. 3b). Motif discovery[12] was performed on each of these groups and revealed that the CTCF motif was the top hit. The distribution of the peaks in each of these groups relative to gene features is shown in Fig. 3c.

For qChIP-seq data spiked in with S2 chromatin, sequences were aligned to both human (hg19) *Drosophila melanogaster* (dm6) genome. Each sample was normalised by the total reads aligned to dm6 (see Active Motif protocol for Spike in Normalisation). Coverage files, average profiles and heatmap were created with Deeptools packages. Quantification of single peaks was calculated with Bamtools package.

### ChIP-qPCR
The ChIP protocol was performed as described in the ChIP-seq session, at the end the DNA was diluted and analysed by qPCR using Mesa Blue mastermix (Eurogentec). Primers were previously published in ref. 22. CTCF site Primer 1 left: GCAAGGCTCTACCGTCATTC, Primer 1 right: CCTTCTCTTCAGAAGCCGTG; Chr:12, 38,787,634-38,787,825. CTCF site Primer 2 (25 in ref. 22) left: CAGCTCTGTGTCCTGTCTTATCC; right: CAGCTATAATTGATGAAGAGGCG; Chr:6, 132,642,584–132,642,808. Non-CTCF site primer (28 in ref. 22) left: GAGCTCTAAGGGAGGCTCCG. Right: CATCATGGTGTCCTCACAGG, Chr:11, 1,983,833–1,983,994.

### RT-qPCR
RNA was extracted using RNeasy plus mini kit Qiagen. Before column purification, cell pellets were vortexed for 30 s in RLT buffer + 1% β-mercaptoethanol. RT-qPCR was carried out using Mesa Blue mastermix (Eurogentec). All reactions were normalised to GAPDH as a control. Primer sequences are listed in Supplementary Data 1.

### Statistics
Statistical significance was analysed using Mann–Whitney test and the Student's *t* test. When appropriate, S-phase cells were defined as the portion of cells where RPA2 > 40 a.u. (Fig. 1f, g), RPA2 > 50 a.u. (Fig. 2a and Supplementary Fig. 3a), RPA2 > 25 a.u. (Supplementary Fig. 1q, r) RPA2 > 18 a.u. (Supplementary Fig. 3b). Figure 1g and Supplementary Fig. 1r: statistical significance was measured using the Mann–Whitney test. Only S-phase cells were analysed, defined as cells where RPA2 > 40 a.u. in Fig. 1g and RPA2 > 25 a.u. in Supplementary 1r. Figures 2a and Supplementary Fig. 3a, 3b: statistical significance was measured using the Mann–Whitney test. Total cells, S-phase cells and G1 phase cells were analysed, as indicated in figure legends. S-phase cells are defined as cells where RPA2 > 50 a.u. or RPA2 > 18 a.u.; G1 phase cells are defined as cells where RPA2 < 50 a.u. or RPA2 < 18 a.u. At least 250 cells were analysed for each experiment. In Fig. 2a and Supplementary 3a, b, values for each experiment were normalised to the mean of that experiment and they were grouped together if the Gaussian distribution was comparable. Supplementary Fig. 2k: statistical significance was measured using the Mann–Whitney test. S-phase cells were defined as the portion of cells where EdU > 8 a.u.; G1 phase cells are defined as cells where EdU < 8 a.u. Supplementary Fig. 2l: statistical significance was measured

using the Mann–Whitney test. At least 250 cells were analysed for each condition in each repeat. Figures 1d, 3d, 3h, 4g, 4k, Supplementary Figs. 1k, 1n, 2c, 2j, 2m, 3e, 3f, 3k, 4e, 4f, 4o: statistical significance was measured using the Mann–Whitney test. Figures 3b, 3f, 4e: statistical significance was measured using the Mann–Whitney test. Supplementary Fig. 1e, 1f, 1h, 3c, 3g, 3i, 3n, 4c, 4d, 4g: statistical significance was measured using the Student's *t* test.

### Reporting summary
Further information on research design is available in the Nature Portfolio Reporting Summary linked to this article.

## Data availability
All data supporting the findings of this study are available within the paper and its Supplementary Information. Any additional information required to reanalyse the data reported in this paper is available from the corresponding authors upon request. Sequencing data have been deposited in GEO under accession code GSE146766 (ChIP SMC1 in RPE1 hTERT c-Myc ER cells treated with 4-hydroxytamoxifen for 48 h or untreated) and GSE249375 (quantitative ChIP SMC1 in RPE1 hTERT c-Myc ER cells treated with 4-hydroxytamoxifen for 48 h or untreated). Source data are provided with this paper.

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

## Acknowledgements
We thank the Crick Institute Advanced Sequencing Facility for the ChIP-seq. We thank Dr. J.-M. Peters for kindly providing GFP-MAU2 plasmid. We would also like to thank to Dr. F. Peri for help with the statistical analysis. This work was supported by core funding to the MRC-UCL University Unit (Ref. MC_EX_G0800785) and funded by R.d.B.'s Cancer Research UK Programme Foundation Award and V.C.'s Wellcome and Royal Society, Sir Henry Dale (221978/Z/20/Z).

## Author contributions
S.P., C.P., V.C., P.T., C.B. and R.d.B. designed research; S.P., H.P., S.R., L.M., K.K. and C.B. performed research; S.P., T.S., C.P. and C.B. analysed the data; and S.P., C.B. and R.d.B. wrote the paper.

## Competing interests
The authors declare no competing interests.
