## [Peer Review File · Nature Communications]

Oncogenic c-Myc induces replication stress by increasing cohesins chromatin occupancy in a CTCF-dependent mannerREVIEWER COMMENTS

Reviewer #1 (Remarks to the Author):

The manuscript "Oncogenic c-Myc induces replication stress by increasing cohesins chromatin occupancy" by Peripolli et al., analyzes Myc induced replicative stress (RS) and shows that increased synthesis of Mau2 leads to higher loading of the cohesion complex on chromatin, which in turn slows DNA-synthesis and increase DDR.

The topic is of high relevance since Myc induce replicative stress is still not fully understood, it follows that further mechanistic insight is certainly needed and critical.

Thus, this work is potentially interesting, yet the evidence presented does not completely support the model proposed.

Also, based on what is shown in the manuscript, it would be more appropriate to present cohesins loading as one of the processes critical for the induction of Myc induced-RS, and not as the main cause since the data included in the manuscript does not allow to rule out the contribution of other processes like for instance Oxidative damage, metabolic interference and transcription-replication conflicts.

Following are the points that will need to be addressed.

Major points:

1. Myc levels (and activity) are critical to reach the so called oncogenic threshold, which is necessary to induced robust gene expression, to alter cell cycle dynamics, trigger apoptosis and replicative stress (doi: 10.1016/j.ccr.2008.10.018). Reaching such threshold is critical when studying the oncogenic properties of Myc. Hallmarks of oncogenic Myc activation in cell culture are the alteration of cell cycle distribution in asynchronous culture conditions and the induction of apoptosis. The authors do not seem to report such effects in their RPE1-MycER cells, which may indicate that in their system ectopic activation of Myc may not have reached the "oncogenic threshold". Can the author provide evidences that indeed MycER activation in their system is indeed above the oncogenic threshold?

2. Checkpoint activation and DDR should be analysed in more details: Is Myc induced replication stress leading to checkpoint activation? Have the authors tried to better characterized the DDR, besides gH2Ax? Are cells positive for gH2Ax showing foci or diffuse staining?

3. Concerning DNA-fiber/Combing experiments:

(i) track length data: please, show the replicates in a supplementary figure.

(ii) Provide the analysis for ongoing fibres, replication origins, replication terminations and stalled forks

(iii) Provide measurements for symmetric, asymmetric and unidirectional forks

4. Mitigation of RS by loss of "cohesins" may represent a way to bypass intrinsic tumor suppression elicited by Myc and favor tumor progression. Thus, testing whether loss of cohesins will enhance Myc induced transformation in-vitro or in-vivo, will greatly improve the relevance of this manuscript. This may explain why cohesins are mutated in tumors.

5. ChIP-seq analyses need to be improved to be fully convincing: a snapshot of a genomic region and a general increase in peaks detection is not sufficient. As such, the data is not strong enough to support the claim that Myc activation promotes cohesins loading. Also, Butchel et al (doi: 10.1016/j.celrep.2017.11.090) proposed that Myc recruits RAD21 on promoters of its target genes, is the increased chromatin association of Smc1 leading to higher binding to Myc target genes?

6. Page 4: Authors claim that "This data also suggests that increased c-Myc-induced transcription is unlikely to be involved in causing replication stress." It is not clear why they conclude this and I think this conclusion is misleading: to my knowledge, the most recent and convincing model that explains

the link between Myc-induced replication stress and transcription was proposed by the Halazonetis lab (doi: 10.1038/nature25507). This model is based on the evidence that Myc activation prior to S-phase entry, accelerates S-phase entry, thus leading to the firing of origins (silent in normal cells) which are mainly positioned on G1-transcribed gene. At these loci, simultaneous DNA replication and transcription, leads to instability of replication forks. It follows that this replication/transcription conflict happens only if Myc is activated prior to S-phase entry, but not if Myc is activated during S-phase. Hence, the evidence reported by the authors cannot fully address whether Myc induction of RS is linked to transcription, since Myc was activated during S-phase (see fig.S2) and not before S-phase entry.

7. The Authors claim that Myc induced RS is due to increase synthesis of Mau2, driven by Myc. Yet, when they activate Myc in S-phase there is no induction of RS, despite there is Myc dependent transcription, as assessed by WB of cycE1, a known Myc target (Fig. S2E). This raises the question of whether Mau2 and more in general other Myc target genes are transcribed as well, or whether Myc, during S-phase only controls the synthesis of a subset of its target genes. This point is critical, as it is expected that, based on the model proposed by the Authors, Mau2 should not increase when Myc is activated in S-phase. A comprehensive gene (possibly by RNA-seq) and protein expression analysis (by WB) is needed.

8. A fundamental corollary of the Authors' model is that over-expression of Mau2 should mimic Myc overexpression and lead to RS. This needs to be verified.

9. Data in figure 3,4 has to be complemented by DDR analysis in order to support the claim of the rescue of replicative stress by cohesins' loss. In the present form, the data only show the rescue of DNA-synthesis, which may not necessarily imply also a rescue of RS.

Minor points:

10. It is reported that loss of RAD21 leads to Myc induced RS (<https://doi.org/10.1038/cddis.2017.345>). This data should be discussed along with the new evidence included in the present manuscript as it appears that too high (this manuscript) or too little cohesins (<https://doi.org/10.1038/cddis.2017.345>) may lead to similar phenotypes (i.e. high RS).

11. It would be fair to properly acknowledge the existing literature linking Myc and cohesins when explaining the rationale for focusing on cohesins as effectors for Myc induced RS and when discussing the results.

Reviewer #2 (Remarks to the Author):

Peripolli et al convincingly demonstrate that expression of oncogenic c-Myc results in cohesin dependent replication stress in human cells. They also demonstrate that the c-Myc induced replication stress in this system does not appear to be linked to dysregulated origin activation or replication transcription collisions. Finally, they show that c-Myc induction leads to increased detection of cohesin complexes on chromosomes and that this is likely linked to overexpression of the cohesin chromatin loading factor Mau2. This provides a plausible pathway for replication stress through excessive loading of cohesin onto chromosomes.

These are novel, intriguing and important results for anybody interested in replication stress and its links to cancer evolution. There are a few minor points of revision that should be incorporated before publication.

1) Although this is the first paper to report that that c-Myc expression results in cohesin dependent replication stress, it is not the first study to show that cohesin dysregulation results in replication stress in mammalian cells. The Losada lab recently showed that loss of Pds5 function increases the chromatin association of cohesin and replication stress in MEFs (Morales C, Ruiz-Torres M, Rodríguez-Acebes S, Lafarga V, Rodríguez-Corsino M, Megías D, Cisneros DA, Peters JM, Méndez J, Losada A. PDS5 proteins are required for proper cohesin dynamics and participate in replication fork protection. *J Biol Chem.* 2020 Jan 3;295(1):146-157. doi: 10.1074/jbc.RA119.011099. Epub 2019 Nov 22. PMID: 31757807; PMCID: PMC6952610).

This study should be quoted either in the results or discussion to demonstrate that there are multiple potential pathways to cohesin causing replication stress in cells.

2) Given the attributed importance of dysregulated origin firing to models of oncogene induced replication stress, I do not think it is enough to just provide a numerical figure for the amount of origin firing occurring in each context in the figures. I think the full analysis of these experiments (i.e. those used to define origin firing) should be outlined in a supplemental figure/table to show how the figures for origin firing were derived from fibre analysis.

3) In the experiments using DRB to deplete transcriptional activity only a visual picture is given in supplementary figure 2H to show that DRB reduces EU incorporation in c-Myc expressing cells. This should be quantified as for gamma H2AX in figure 1e. This will provide a better comparison with the data from reference 7, which argued that replication stress caused by overexpression of cyclin E is attenuated by transcription shutoff via cordycepsin treatment.

Typographic errors

L173 "Increase of cohesions" should be changed to "increase of cohesins"

Reviewer #3 (Remarks to the Author):

In the manuscript titled: 'Oncogenic c-Myc induces replication stress by increasing cohesins chromatin occupancy' the authors have analysed the effects of Myc over expression in RPE1 hTERT cell line and determined increased cohesion chromatin occupancy is the main cause of replication stress. Although the authors have, in the most part, produced a well written and concise study, I have numerous issues with this study, which in its current state I cannot recommend for publication. The effects of RAD21, CTCF and Mau2 knockdown, on Replicative track length, are convincing, novel and intriguing, however at times I am not convinced by the authors conclusions and the study lacks appropriate controls and essential experiments. I feel this study is incomplete and unsuited for a short communication.

Below I outline my concerns.

Major concerns:

1. The authors only provide eIF4E RT-qPCR data set (Supp Fig 1A) as means to demonstrate Myc induction/overexpression. This is insufficient evidence that myc is over expressed and functional. Why weren't westerns or RT-qPCR for myc performed? Without proper controls it reduces confidence in all subsequent experiments. Simply referencing a previous study is also not evidence. The authors must characterise the activation of myc and its functionality using WB, PCR and Immunofluorescence.
2. On lines 111-112 the authors state: 'Our data indicates that c-Myc-induced replication stress depends on events that take place in the G1 phase of the cell cycle. This statement is derived from the HU experiment, which is unpersuasive. The use of hydroxyurea (HU) to synchronize cells and study the effects of myc overexpression in s-phase do not support the conclusion that: 'c-Myc activity during S phase does not cause replication stress.' It is widely known Hydroxyurea (HU) depletes the cells of

dNTPs and causes replication stress (PMC2958316). This stress is clearly demonstrated in Supplementary Figure 2d. MYC on the other hand increases nucleotide biosynthesis and replication initiation events. Therefore, could the increase in fibre length affect myc activation relative to the controls (supp. Fig 2C) be a consequence of myc-induced increased dNTP and/or increased replication events? Examining myc-induced replication stress in cells already experiencing replication stress may well mask or alter some effects of myc-activation. The effects of myc-activation G1 and S-phase would be better assessed by other means, e.g. timed release after contact inhibition.

3. In supplementary figure 2G and 2H, the authors examine the consequence of DRB treatment on transcription-replication conflicts. The images of 5-ethynyl uridine incorporation into RNA suggests DRB reduces EU intensity. However, the authors have failed to demonstrate increased RNA synthesis following myc activation and therefore the validity of this experiment. Quantification of EU/RNA intensity +/- myc and +/- DRB are essential to eliminate transcriptional-replication conflicts and to help demonstrate myc-ER is functional.

4. On lines 175-177 the authors write: 'In all knock-down experiments, the extent of replication origin firing and the cell cycle profiles are not affected by Rad21 depletion (Supplementary fig. 3e and f), suggesting that origin firing and reduced G1 length are unlikely to be involved in c-Myc-induced replication stress.' The authors have not demonstrated or discussed myc-induced reduced G1 length. How have the authors reached this conclusion?

5. In Supplementary Figure 2F the authors examined the mean intensity of MCM7 before and after myc activation. Assessing MCM7 intensity using immunofluorescence does not demonstrate MCM7 binding to chromatin nor is it a sufficient measure of origin licensing. This protocol only shows protein levels in the nucleus. In order to demonstrate chromatin binding the author must perform additional experiments examining chromatin, e.g. Immunoprecipitation, Western Blot or ChIP-seq. This is especially true for the immunofluorescent quantification of SMC1 and SMC3 throughout the manuscript. Soluble SMC1 and SMC3 can be present in the nucleus unbound to chromatin (PMC3680458 and doi: 10.1016/s0092-8674(03)00162-4), providing a reservoir of cohesin.

6. The authors proclaim knock-down of MAU2, RAD21 or CTCF, followed by the subsequent immunofluorescent quantification of SMC1 or SMC3 is sufficient to examine the effects of c-myc-induced changes in Cohesin occupancy. On lines 181-183 the authors write: 'These data indicate that c-Myc-induced accumulation of cohesins at CTCF sites causes a slowdown of replication forks, generating replication stress.' Yet the authors do not examine the effects of MAU2, RAD21 or CTCF knockdown on SMC1 enrichment/occupancy at the genomic level. For example what happens to the 12,390 myc-induced SMC1 peaks after knockdown? Global or site specific changes? At the bare minimum this study requires the examination of SMC1 levels using ChIP-seq following the knockdown of MAU2 (at least) to demonstrate any link between cohesin levels and DNA replication stress.

7. The knock-down or depletion of RAD21 and CTCF (and cohesin) can differentially affect chromatin architecture and gene expression in human cells (PMID: 24335803). The authors have not controlled for this or discuss this in any detail. For example, what are the effects of myc-activation on gene expression or cell cycle profile following RAD21, CTCF or MAU2 knockdown?

8. The manuscript lacks adequate discussion and context. A previous study (PMID: 28749464) proposed RAD21 knockdown induces transcriptional and cell cycle changes. Moreover, RAD21 promotes cell survival upon ectopic activation of Myc: RAD21 knockdown causes replicative stress upon ectopic activation of Myc. This publication is not acknowledged or discussed by this study. On lines 170-172 it reads: 'Reducing the levels of cohesin (RAD21 knockdown) chromatin occupancy in cells experiencing oncogenic c-Myc increases DNA fibre length in the first S-phase in both synchronised and asynchronous cell populations (Fig. 3c and d, Supplementary fig. 3d).' How do the authors account the discrepancies between their study and published data? Have the authors assess the impacts of RAD21 knockdown on cell death, cell cycle and/or transcription?

9. In the conclusion on lines 214-216 it reads: 'Overall our data show that excessive cohesins on chromatin can interfere with the progression of replication forks, thus contributing to replication stress.' The authors have not provided any mechanistic evidence that increased cohesin directly interferes with the progression of replication forks. How have they reached this conclusion? Can overexpression of SMC1, SMC3, RAD21 or MAU2 cause replication stress? Testing the effects of SMC1, SMC3, RAD21 or MAU2 overexpression in Retinal Pigment Epithelia 1 (ATCC CRL-4000) ER empty cells

would provide some mechanistic evidence for cohesin induced replication stress.

Minor concerns:

1. Is Fig 1a necessary? It does not complement or clarify any information already provided in the text.
2. Line 42-43 of introduction: 'In the case of oncogenic overexpression of Cyclin E, both mechanisms have been reported.' The preceding sentences do not justify the use of the word 'both' in this sentence.
3. How were the 'Fractions of cells in different phases of the cell cycle' in Supplementary Figures 1B determined? This cannot be performed with PI alone and would require PI and EdU/BrdU Flow cytometry. How were the levels of fluorescent intensity determined for G1 and S-phase cells in Supp Figure 3A&B, Figure 2a? Additional information in the Materials and Methods is required.
4. On lines 107-109 the author states: 'This data also suggests that increased c-Myc-induced transcription is unlikely to be involved in causing replication stress.' Up until that point of the manuscript the authors have not provided any evidence for the absence of replication-transcription conflicts following myc-activation.
5. On lines 135-136 the authors write: 'We hypothesised that, upon c-Myc activation, an increase in cohesin occupancy could slow down replisome progression during S phase.' Why? There are countless events occurring in G1-phase that could cause reduced fork speeds. Why cohesin, why not CTCF?
6. In western blots (Figure 2b) what does TOTAL indicate? If this indicates total cell lysates why is Histone H3 detected/assayed?
7. The ChIP-seq data-set looks good! Was the SMC1 antibody ChIP-grade? Images from the manufacturer's website would suggest otherwise. How have the author validated this antibody for ChIP-seq? The accession number should be provided in the methods. This analysis would benefit from some profile plots or heat maps looking at the levels/enrichment of SMC1 at CTCF sites. Are SMC1 levels increased at pre-existing sites or is the significant increase only at novel sites?
8. Why is Supplementary Figure 4A, examining the levels of RAD21 and SMC1, included after numerous experiments examining the effects of RAD21 and CTCF knockdown. These should be included earlier in the manuscript.
9. The authors need to check their statistical analysis. A few figures (e.g. Fig 4e) averages and SEMs between treatments are almost identical however the stats suggest the difference is highly significant. Moreover, the authors have failed to describe how the intensity of SMC1, SMC3, H2AX etc are quantified. How many nuclei are counted? How many replicates? Are sample blinded before counting? how have the authors controlled for differences in fluorescent intensity between experiments?
10. Reference 32 (main references) is incorrect and requires updating.

REVIEWER COMMENTS

AUTHOR REBUTTAL

Reviewer #1 (Remarks to the Author):

The manuscript "Oncogenic c-Myc induces replication stress by increasing cohesins chromatin occupancy" by Peripolli et al., analyzes Myc induced replicative stress (RS) and shows that increased synthesis of Mau2 leads to higher loading of the cohesion complex on chromatin, which in turn slows DNA-synthesis and increase DDR. The topic is of high relevance since Myc induce replicative stress is still not fully understood, it follows that further mechanistic insight is certainly needed and critical. Thus, this work is potentially interesting, yet the evidence presented does not completely support the model proposed.

Also, based on what is shown in the manuscript, it would be more appropriate to present cohesins loading as one of the processes critical for the induction of Myc induced-RS, and not as the main cause since the data included in the manuscript does not allow to rule out the contribution of other processes like for instance Oxidative damage, metabolic interference and transcription-replication conflicts. Following are the points that will need to be addressed.

We thank the reviewer for acknowledging that the topic of our work is of high relevance and our findings potentially interesting. We agree with their overall assessment that cohesin loading should be seen as one of the processes critical for the induction of Myc induced-RS and have made changes to the text to better reflect this.

Major points:

1. Myc levels (and activity) are critical to reach the so called oncogenic threshold, which is necessary to induced robust gene expression, to alter cell cycle dynamics, trigger apoptosis and replicative stress (doi: 10.1016/j.ccr.2008.10.018). Reaching such threshold is critical when studying the oncogenic properties of Myc. Hallmarks of oncogenic Myc activation in cell culture are the alteration of cell cycle distribution in asynchronous culture conditions and the induction of apoptosis. The authors do not seem to report such effects in their RPE1-MycER cells, which may indicate that in their system ectopic activation of Myc may not have reached the "oncogenic threshold". Can the author provide evidences that indeed MycER activation in their system is indeed above the oncogenic threshold?

We thank the reviewer for highlighting this important point. Data establishing the hallmarks of oncogenic Myc activation for our c-Myc inducible system, listed by the reviewer, were presented but maybe not explicitly referred to, as such, in the text. In the revised manuscript we have clearly referred to this data as hallmarks of oncogenic Myc activation for our c-Myc inducible system and included additional data to further strengthen this point. Specifically, the effect of oncogenic Myc activation on cell cycle distribution in asynchronous cells at 24 and 48h after induction is shown in supplementary figure 1h and colony formation as a measure of cell death (Supplementary 1d), we show the levels of ER-cMyc protein in western Blot and IF (Supplementary fig. 1a,b and c) and DNA damage checkpoint activation and replication stress (Supplementary 1i, j and k). We also added data on the transcriptional induction of additional well-known Myc target genes (Supplementary 1e, f and g) and more clearly refer to the characterization of this system, by colony formation assay and western blot analysis, in our previously published paper, Bertoli et al. 2016.

2. Checkpoint activation and DDR should be analysed in more details: Is Myc induced replication stress leading to checkpoint activation? Have the authors tried to better characterized the DDR, besides gH2Ax? Are cells positive for gH2Ax showing foci or diffuse staining?

We have now included checkpoint-dependent phosphorylation of RPA, Chk1 and Chk2 by western blot analysis, that indicates activation of the replication stress and DNA damage checkpoint response (Supplementary 1j). Based on the data reported in fig 1e-f and supplementary 1o-q, showing RPA and gammaH2AX in extracted nuclei, the staining in gamma-positive cells is mainly in foci, with only a few cells showing a more diffuse staining.

3. Concerning DNA-fiber/Combing experiments:

- (i) track length data: please, show the replicates in a supplementary figure.
- (ii) Provide the analysis for ongoing fibres, replication origins, replication terminations and stalled forks
- (iii) Provide measurements for symmetric, asymmetric and unidirectional forks

We now include the requested data concerning DNA-fibre experiments (listed below) in the new version (Supplementary fig. 1l). While the length of DNA fibers is significantly reduced in all assays, the ratio between green and red tracks, indicative of fork asymmetry, does not change significantly, therefore we used fiber length as a measure of RS throughout the manuscript.

- (i) We have added the replicates of the DNA fibers assay as requested.

(ii) The data of ongoing fibers and origin firing are already present, we now reported the firing data as separate graphs.
(iii) The data on fork symmetry/asymmetry have been added.

4. Mitigation of RS by loss of “cohesins” may represent a way to bypass intrinsic tumor suppression elicited by Myc and favor tumor progression. Thus, testing whether loss of cohesins will enhance Myc induced transformation *in-vitro* or *in-vivo*, will greatly improve the relevance of this manuscript. This may explain why cohesins are mutated in tumors.

We agree with the reviewer that our findings may explain why cohesins are mutated in tumors. We have now included new data, performed in lung cancer cells with different levels of c-Myc overexpression, that provide support for the reviewer’s point that loss of cohesins in cancer cells should mitigate RS elicited by c-Myc (Figure 4h-k). Whilst we agree that establish the role of cohesins in tumorigenesis *in vitro* and most certainly *in vivo*, as suggested by the reviewer, would provide further relevance of our findings, we feel that this is beyond the scope of this manuscript.

5. ChIP-seq analyses need to be improved to be fully convincing: a snapshot of a genomic region and a general increase in peaks detection is not sufficient. As such, the data is not strong enough to support the claim that Myc activation promotes cohesins loading.

We agree with the reviewer that ChIP-seq data is not a robust quantitative assessment of chromatin occupancy. Whilst the ChIP-seq data does suggest a general increase at peaks, as pointed out by the reviewer, in our paper this data is mainly used to reveal significant enrichment at many additional (mostly CTCF) sites. Our conclusion, that Myc activation promotes cohesin loading, is based on quantitative IF on extracted nuclei (Fig 2a and suppl 3a,b) and chromatin extraction followed by western blot (Fig 2b) to quantify Smc1 and Smc3 binding to chromatin. These data do support our claim that Myc activation promotes an increase in chromatin occupancy. Importantly, we also show that reducing the levels of the cohesin subunit Rad21 reduces the levels of Smc1 on DNA in quantitative IF, which validates our IF data (Fig 3 a,b). The ChIP-seq data is therefore mainly used as a qualitative, rather than a quantitative, assessment of where the additional cohesins end up.

Also, Butchel et al (doi: 10.1016/j.celrep.2017.11.090) proposed that Myc recruits RAD21 on promoters of its target genes, is the increased chromatin association of Smc1 leading to higher binding to Myc target genes?

Unfortunately, as pointed out by the reviewer, the ChIPseq data is not sufficiently quantitative to establish an increase in peak detection of Rad21 at c-Myc target promoters. However, to test the possibility that a potential increase in Rad21 chromatin occupancy at c-Myc targets could affect c-Myc’s ability to activate its target genes we have analysed the mRNA levels of some c-Myc targets in control and Rad21 depleted cells. We do not detect a significant difference in mRNA levels between control and Rad21 depleted cells (Supplementary 3h).

6. Page 4: Authors claim that “This data also suggests that increased c-Myc-induced transcription is unlikely to be involved in causing replication stress.” It is not clear why they conclude this and I think this conclusion is misleading: to my knowledge, the most recent and convincing model that explains the link between Myc-induced replication stress and transcription was proposed by the Halazonetis lab (doi: 10.1038/nature25507). This model is based on the evidence that Myc activation prior to S-phase entry, accelerates S-phase entry, thus leading to the firing of origins (silent in normal cells) which are mainly positioned on G1-transcribed gene. At these loci, simultaneous DNA replication and transcription, leads to instability of replication forks. It follows that this replication/transcription conflict happens only if Myc is activated prior to S-phase entry, but not if Myc is activated during S-phase. Hence, the evidence reported by the authors cannot fully address whether Myc induction of RS is linked to transcription, since Myc was activated during S-phase (see fig.S2) and not before S-phase entry.

We agree with the reviewer that we cannot exclude a contribution of transcription-replication collisions based on the evidence we present. We have now made sure to emphasise this and that cohesin loading should be seen as one of the processes critical for the induction of Myc-induced-RS and have made changes to the text to better reflect this.

7. The Authors claim that Myc induced RS is due to increase synthesis of Mau2, driven by Myc. Yet, when they activate Myc in S-phase there is no induction of RS, despite there is Myc dependent transcription, as assessed by WB of *cycE1*, a known Myc target (Fig. S2E). This raises the question of whether Mau2 and more in general other Myc target genes are transcribed as well, or whether Myc, during S-phase only controls the synthesis of a subset of its target genes. This point is critical, as it is expected that, based on the model proposed by the Authors, Mau2 should not increase when Myc is activated in S-phase. A comprehensive gene (possibly by RNA-seq) and protein expression analysis (by WB) is needed.

We would like to thank the reviewer for pointing this out and realize we should explain our understanding of the cell cycle-dependent role of Mau2 in cohesin regulation better to clarify our rationale for its potential involvement in Myc-induced RS. In short, cohesin is loaded onto the DNA after mitosis and during G1 in a Mau2-dependent

manner, but its association with chromatin is mostly dynamic. Only a subset of these cohesins, together with CTCF, are more stably interacting with the DNA to establish the three-dimensional structure of chromatin, which happens in G1. Our data suggests that it is this particular subset of cohesins/CTCF that increases in a c-Myc-dependent manner, which is at the likely basis of causing RS in the subsequent S phase. Cells released directly into S phase with myc overexpression, already determined the cohesins/CTCF interactions during the previous G1, which remains largely unaffected during S phase where cohesin establishment has a more dominant role in stabilizing cohesin/DNA interactions. Based on this, whether Mau2 does or does not increase in S phase in a c-Myc-dependent manner, it is unlikely to affect cohesins/CTCF interactions, which are established in G1.

Following the reviewer's suggestion, we have analysed the levels of Mau2 in the HU release experiment and detect a small increase after 8 hours of c-Myc activation, which corresponds to the late S mostly and is slightly later than the cyclinE induction (Supplementary 2f). It is therefore possible that there is a difference between c-Myc-dependent G1 and S transcription and we have added a sentence in the paper to acknowledge this.

8. A fundamental corollary of the Authors' model is that over-expression of Mau2 should mimic Myc overexpression and lead to RS. This needs to be verified.

We agree that this is an important experiment and now include data that shows that overexpressing Mau2 alone, in transient transfection in RPE1 cell, causes an increase in phospho-RPA and gammaH2AX via western blot and IF (Supplementary 4h-m). These data support our hypothesis that an increase in Mau2-dependent cohesin loading could contribute to c-Myc-induced RS.

9. Data in figure 3,4 has to be complemented by DDR analysis in order to support the claim of the rescue of replicative stress by cohesins' loss. In the present form, the data only show the rescue of DNA-synthesis, which may not necessarily imply also a rescue of RS.

The definition for RS we use in the paper is - slowing down and/or stalling or replication forks. The DNA fibre analysis, used as a proxy for replication fork speed, shows that preventing an increase in cohesin chromatin occupancy in cells experiencing oncogenic c-Myc activity increases fibre length, rescuing slowing down of replication forks and therefore, a rescue of RS. We do agree that a decrease in DDR could also indicate a rescue of RS, but indirectly via a decrease in RS-induced DNA damage, so we have now added DDR analysis in the supplementary data (Supplementary 3l-m).

Minor points:

10. It is reported that loss of RAD21 leads to Myc induced RS (<https://doi.org/10.1038/cddis.2017.345>). This data should be discussed along with the new evidence included in the present manuscript as it appears that too high (this manuscript) or too little cohesins (<https://doi.org/10.1038/cddis.2017.345>) may lead to similar phenotypes (i.e. high RS).

We agree with the reviewer that this is an important point. We now highlight, in the discussion, this seemingly dual role of cohesin in causing c-Myc-induced RS and its previously reported role preventing RS-induced DNA damage.

11. It would be fair to properly acknowledge the existing literature linking Myc and cohesins when explaining the rationale for focusing on cohesins as effectors for Myc induced RS and when discussing the results.

The existing literature suggests an important role for cohesins in the response to RS, specifically in the repair of RS-induced DNA damage. Its role in a mechanism that causes c-Myc-induced RS, reported here, is the complete opposite of this and was therefore, if anything, a reason not to look at cohesins. As mentioned above, we now highlight, in the discussion, this seemingly dual role of cohesin in causing c-Myc-induced RS and its previously reported role in preventing RS-induced DNA damage.

Reviewer #2 (Remarks to the Author):

Peripolli et al convincingly demonstrate that expression of oncogenic c-Myc results in cohesin dependent replication stress in human cells. They also demonstrate that the c-Myc induced replication stress in this system does not appear to be linked to dysregulated origin activation or replication transcription collisions. Finally, they show that c-Myc induction leads to increased detection of cohesion complexes on chromosomes and that this is likely linked to overexpression of the cohesin chromatin loading factor Mau2. This provides a plausible pathway for replication stress through excessive loading of cohesin onto chromosomes.

These are novel, intriguing and important results for anybody interested in replication stress and its links to

cancer evolution. There are a few minor points of revision that should be incorporated before publication.

We thank the reviewer for their positive assessment of our work and their conclusion that these are novel, intriguing and important results.

1) Although this is the first paper to report that that c-Myc expression results in cohesin dependent replication stress, it is not the first study to show that cohesin dysregulation results in replication stress in mammalian cells. The Losada lab recently showed that loss of Pds5 function increases the chromatin association of cohesin and replication stress in MEFs (Morales C, Ruiz-Torres M, Rodríguez-Acebes S, Lafarga V, Rodríguez-Corsino M, Megías D, Cisneros DA, Peters JM, Méndez J, Losada A. PDS5 proteins are required for proper cohesin dynamics and participate in replication fork protection. *J Biol Chem.* 2020 Jan 3;295(1):146-157. doi: 10.1074/jbc.RA119.011099. Epub 2019 Nov 22. PMID: 31757807; PMCID: PMC6952610). This study should be quoted either in the results or discussion to demonstrate that there are multiple potential pathways to cohesin causing replication stress in cells.

We would like to thank the reviewer for pointing out this study, which complements our study into the role of cohesin regulation in oncogene-induced replication stress. We now refer to this study in our discussion.

2) Given the attributed importance of dysregulated origin firing to models of oncogene induced replication stress, I do not think it is enough to just provide a numerical figure for the amount of origin firing occurring in each context in the figures. I think the full analysis of these experiments (i.e. those used to define origin firing) should be outlined in a supplemental figure/table to show how the figures for origin firing were derived from fibre analysis.

We now have included a table including the origin firing data (Supplementary fig 11), in addition to the graphs present in supplementary fig. 3f and 4g.

3) In the experiments using DRB to deplete transcriptional activity only a visual picture is given in supplementary figure 2H to show that DRB reduces EU incorporation in c-Myc expressing cells. This should be quantified as for gamma H2AX in figure 1e. This will provide a better comparison with the data from reference 7, which argued that replication stress caused by overexpression of cyclin E is attenuated by transcription shutoff via cordycepsin treatment.

We now have included a graph for the quantification of the EU incorporation (Supplementary fig. 2m).

Typographic errors

L173 "Increase of cohesions" should be changed to "increase of cohesins"

We have now corrected this typo.

Reviewer #3 (Remarks to the Author):

In the manuscript titled: 'Oncogenic c-Myc induces replication stress by increasing cohesins chromatin occupancy' the authors have analysed the effects of Myc over expression in RPE1 hTERT cell line and determined increased cohesion chromatin occupancy is the main cause of replication stress. Although the authors have, in the most part, produced a well written and concise study, I have numerous issues with this study, which in its current state I cannot recommend for publication. The effects of RAD21, CTCF and Mau2 knockdown, on Replicative track length, are convincing, novel and intriguing, however at times I am not convinced by the authors conclusions and the study lacks appropriate controls and essential experiments. I feel this study is incomplete and unsuited for a short communication.

We thank the reviewer for their assessment that we produced a well written and concise study and that important pieces of data are convincing, novel and intriguing.

Below I outline my concerns.

Major concerns:

1. The authors only provide eIF4E RT-qPCR data set (Supp Fig 1A) as means to demonstrate Myc induction/overexpression. This is insufficient evidence that myc is over expressed and functional. Why weren't westerns or RT-qPCR for myc performed? Without proper controls it reduces confidence in all subsequent experiments. Simply referencing a previous study is also not evidence. The authors must characterise the activation of myc and its functionality using WB, PCR and Immunofluorescence.

We would like to thank the reviewer for pointing out the need to explain our c-Myc inducible system in more detail. In short, activation of c-Myc in our c-Myc-ER inducible cell line, used in this work and characterized in published work (Bertoli et al 2016), is based on release from chaperones and translocation of the c-Myc-ER protein into the nucleus, upon 4-OH addition, rather than c-Myc transcriptional induction/expression. Therefore

western or RT-qPCR analysis for c-Myc in this cellular system is not indicative of c-Myc activity. The cell line was developed and characterized in our study published in 2016, referred to in the text, which does include western blot analysis to show cMyc-ER protein levels, but more importantly the effects of c-Myc activation on replication stress-induced DNA damage by single cell IF of RPA (RS) and gamma-H2AX (DNA damage) and colony formation. In the current study we further characterized the effect on 1) the well-established c-Myc target eIF4E by RT-qPCR, 2) cell cycle profile, and 3) the replication stress and DNA damage phenotypes (Supplementary 1). For completeness, we have now added the levels of c-Myc ER in WB (Supplementary 1c), additional c-Myc transcriptional targets (Supplementary 1f and g), a colony survival assay (Supplementary 1d) and the IF showing the translocation of the protein to the nucleus upon 4OHT addition (Supplementary 1a and b).

2. On lines 111-112 the authors state: 'Our data indicates that c-Myc-induced replication stress depends on events that take place in the G1 phase of the cell cycle. This statement is derived from the HU experiment, which is unpersuasive. The use of hydroxyurea (HU) to synchronize cells and study the effects of myc overexpression in s-phase do not support the conclusion that: 'c-Myc activity during S phase does not cause replication stress.' It is widely known Hydroxyurea (HU) depletes the cells of dNTPs and causes replication stress (PMC2958316). This stress is clearly demonstrated in Supplementary Figure 2d. MYC on the other hand increases nucleotide biosynthesis and replication initiation events. Therefore, could the increase in fibre length affect myc activation relative to the controls (supp. Fig 2C) be a consequence of myc-induced increased dNTP and/or increased replication events? Examining myc-induced replication stress in cells already experiencing replication stress may well mask or alter some effects of myc-activation. The effects of myc-activation G1 and S-phase would be better assessed by other means, e.g. timed release after contact inhibition.

We agree with the reviewer that most treatments widely used to synchronize cells in S phase, which includes HU, do so by causing RS and are therefore not ideal when studying oncogene-induced RS. To circumvent this issue, we have now added a G1 arrest and release experiment using the CDK4-6 inhibition Palbociclib, where c-Myc is activated soon after release, in G1, or just prior to entering S phase. WB and DNA fibre analysis in early and middle/late S phase (supplementary 2i and j) confirm the observation made upon release from S-phase by HU and G1 by contact inhibition.

3. In supplementary figure 2G and 2H, the authors examine the consequence of DRB treatment on transcription-replication conflicts. The images of 5-ethynyl uridine incorporation into RNA suggests DRB reduces EU intensity. However, the authors have failed to demonstrate increased RNA synthesis following myc activation and therefore the validity of this experiment. Quantification of EU/RNA intensity +/- myc and +/- DRB are essential to eliminate transcriptional-replication conflicts and to help demonstrate myc-ER is functional.

We quantified the levels of EU incorporation and confirmed that c-Myc increases global transcription levels and that DRB decreases it (Supplementary 2m).

4. On lines 175-177 the authors write: 'In all knock-down experiments, the extent of replication origin firing and the cell cycle profiles are not affected by Rad21 depletion (Supplementary fig. 3e and f), suggesting that origin firing and reduced G1 length are unlikely to be involved in c-Myc-induced replication stress.' The authors have not demonstrated or discussed myc-induced reduced G1 length. How have the authors reached this conclusion?

We would like to thank the reviewer for pointing this out. We carried out an EdU incorporation experiments, counting the percentage of EdU positive cells at various timepoints after release from confluency. We observe c-Myc-induced cells start to synthesise DNA approximately 2h before control cells, indicating a reduced G1 length (data not shown). This is in line with the reduced G1 length upon c-Myc activation in synchronized c-Myc U2OS observed previously (M Macheret and T D Halazonetis, Nature 2018). A reduction in G1 length is thought to potentially cause RS via reduced licensing (Ekholm-Reed et al, 2004). However, whilst we see a reduction in G1 length we do not observe a reduction in origin licensing suggesting that this is unlikely to be involved in c-Myc-induced replication stress.

5. In Supplementary Figure 2F the authors examined the mean intensity of MCM7 before and after myc activation. Assessing MCM7 intensity using immunofluorescence does not demonstrate MCM7 binding to chromatin nor is it a sufficient measure of origin licensing. This protocol only shows protein levels in the nucleus. In order to demonstrate chromatin binding the author must perform additional experiments examining chromatin, e.g. Immunoprecipitation, Western Blot or ChIP-seq. This especially true for the immunofluorescent quantification of SMC1 and SMC3 throughout the manuscript. Soluble SMC1 and SMC3 can be present in the nucleus unbound to chromatin (PMC3680458 and doi: 10.1016/s0092-8674(03)00162-4), providing a reservoir of cohesin.

In all our IF experiments we pre-extract the nuclear proteins before fixation, so only chromatin-bound proteins remain. We measured the levels of chromatin-bound MCM7, Smc1 and Smc3 throughout the manuscript. We thank the reviewer for alerting us that this was unclear and have made sure to point this out early on in the text to prevent confusion.

6. The authors proclaim knock-down of MAU2, RAD21 or CTCF, followed by the subsequent immunofluorescent quantification of SMC1 or SMC3 is sufficient to examine the effects of c-myc-induced changes in Cohesin occupancy. On lines 181-183 the authors write: 'These data indicate that c-Myc-induced accumulation of cohesins at CTCF sites causes a slowdown of replication forks, generating replication stress.' Yet the authors do not examine the effects of MAU2, RAD21 or CTCF knockdown on SMC1 enrichment/occupancy at the genomic level. For example what happens to the 12,390 myc-induced SMC1 peaks after knockdown? Global or site specific changes? At the bare minimum this study requires the examination of SMC1 levels using ChIP-seq following the knockdown of MAU2 (at least) to demonstrate any link between cohesin levels and DNA replication stress.

We used the quantitative IF to test the levels of global cohesins on DNA upon myc induction and following Rad21, Mau2 and CTCF depletion. Upon Rad21 and Mau2 depletion we observed a reduction in the global levels of SMC1 on DNA, confirming the published data and supporting the validity of this approach. Moreover, the ChIPseq data are not the best quantitative assessment of interactions with the DNA.

Our quantitative chromatin IF shows that c-Myc activation causes an overall increase in chromatin-associated cohesin, and our ChIP-seq data shows accumulation at additional CTCF sites. This suggests that the increased chromatin associated cohesin accumulated at CTCF sites, which is likely the cause of RS. However, we agree with the reviewer that even though our IF data shows that a global reduction of chromatin-bound cohesins, via Rad21 and Mau2 knockdown, rescues slowing down of replication forks it does not assess the loss of accumulation at additional CTCF sites. We have therefore made changes to text to reflect this. The sentence "These data indicate that c-Myc-induced accumulation of cohesins at CTCF sites causes a slowdown of replication forks, generating replication stress" has been changed to "These data indicate that c-Myc-induced increased cohesins chromatin occupancy, likely at CTCF sites, causes a slowdown of replication forks, generating replication stress".

7. The knock-down or depletion of RAD21 and CTCF (and cohesin) can differentially affect chromatin architecture and gene expression in human cells (PMID: 24335803). The authors have not controlled for this or discuss this in any detail. For example, what are the effects of myc-activation on gene expression or cell cycle profile following RAD21, CTCF or MAU2 knockdown?

We would like to direct the reviewer to the cell cycle profile data in the supplementary figure 3. In addition, we also present data on origin firing, to assess if this was modified by the depletions. In all case the cell cycle profiles and the firing frequency are similar.

8. The manuscript lacks adequate discussion and context. A previous study (PMID: 28749464) proposed RAD21 knockdown induces transcriptional and cell cycle changes. Moreover, RAD21 promotes cell survival upon ectopic activation of Myc: RAD21 knockdown causes replicative stress upon ectopic activation of Myc. This publication is not acknowledged or discussed by this study. On lines 170-172 it reads: 'Reducing the levels of cohesin (RAD21 knockdown) chromatin occupancy in cells experiencing oncogenic c-Myc increases DNA fibre length in the first S-phase in both synchronised and asynchronous cell populations (Fig. 3c and d, Supplementary fig. 3d).' How do the authors account the discrepancies between their study and published data? Have the authors assess the impacts of RAD21 knockdown on cell death, cell cycle and/or transcription?

We agree with the reviewer and will reference the work they mention. However, we did already acknowledge this discrepancy in the discussion, however briefly due to word limit. We mentioned that previous reports support a role for Rad21 in protecting stalled forks and repairing DNA damage, which is the opposite from what we see. An important point that could explain this difference is that we purposefully use an inefficient siRNA, to merely prevent c-Myc from increasing cohesin on chromatin. We did this to prevent reducing cohesins levels entirely exactly for the reason that this has been shown to cause RS. Another crucial difference is that we are looking in the first S phase, when loss of Rad21 could not affect events such as mitosis, since mitotic defect could *per se* generate RS in the following S phase. The study mentioned, which has now been referenced, detects DNA damage and RS at 48h and 72h post transfection, while we study RS within the first 24h, and in the case of synchronized cells in the very first S phase, before any mitosis had taken place. We have now extended our discussion to reflect on the discrepancy between our results and those published and conclude that cohesin chromatin occupancy can contribute both to the generation and tolerance to RS upon c-Myc activation.

9. In the conclusion on lines 214-216 it reads: 'Overall our data show that excessive cohesins on chromatin can interfere with the progression of replication forks, thus contributing to replication stress.' The authors have not provided any mechanistic evidence that increased cohesin directly interferes with the progression of replication forks. How have they reached this conclusion? Can overexpression of SMC1, SMC3, RAD21 or MAU2 cause replication stress? Testing the effects of SMC1, SMC3, RAD21 or MAU2 overexpression in Retinal Pigment Epithelia 1 (ATCC CRL-4000) ER empty cells would provide some mechanistic evidence for cohesin induced replication stress.

We performed experiments overexpressing Mau2 in RPE1 cells and could detect increase in phospho-RPA in western blot and phospho-RPA and gamma H2AX in IF in Mau2 transfected cells (Supplementary 4h-m). These

new data are included in supplementary and support the idea that an increase in cohesins on DNA can lead to RS.

Minor concerns:

1. Is Fig 1a necessary? It does not complement or clarify any information already provided in the text.

We used it as a visual background information for the role of RS in tumorigenesis for the general reader, correlated to the opening paragraph in the abstract where we say that Rs has a role in cancer initiation and progression.

2. Line 42-43 of introduction: 'In the case of oncogenic overexpression of Cyclin E, both mechanisms have been reported.' The preceding sentences do not justify the use of the word 'both' in this sentence.

We have now corrected the sentence.

3. How were the 'Fractions of cells in different phases of the cell cycle' in Supplementary Figures 1B determined? This cannot be performed with PI alone and would require PI and EdU/BrdU Flow cytometry. How were the levels of fluorescent intensity determined for G1 and S-phase cells in Supp Figure 3A&B, Figure 2a? Additional information in the Materials and Methods is required.

In supplementary 1b, we assessed cell cycle distribution via PI staining, it is not a perfect assessment as pointed out by the reviewer, but it is widely used to visualize cell cycle general differences.

In suppl. fig. 3 a,b we co-stained cells for RPA2 to separate cells in G1 and S phase. In pre-extracted samples, nuclear staining with RPA2 is present only in S phase. We have added this information to the text. How cells were separated between RPA2 positive or negative in each experiment is described in MMs in the statistical analysis section.

4. On lines 107-109 the author states: 'This data also suggests that increased c-Myc-induced transcription is unlikely to be involved in causing replication stress.' Up until that point of the manuscript the authors have not provided any evidence for the absence of replication-transcription conflicts flowing myc-activation.

We have now removed this sentence.

5. On lines 135-136 the authors write: 'We hypothesised that, upon c-Myc activation, an increase in cohesin occupancy could slow down replisome progression during S phase.' Why? There are countless events occurring in G1-phase that could cause reduced fork speeds. Why cohesin, why not CTCF?

Because we had some indications derived from other lines of work in the lab pointing to this.

6. In western blots (Figure 2b) what does TOTAL indicate? If this indicates total cell lysates why is Histone H3 detected/assayed?

We have now removed it.

7. The ChIP-seq data-set looks good! Was the SMC1 antibody ChIP-grade? Images from the manufacturer's website would suggest otherwise. How have the author validated this antibody for ChIP-seq? The accession number should be provided in the methods. This analysis would benefit from some profile plots or heat maps looking at the levels/enrichment of SMC1 at CTCF sites. Are SMC1 levels increased at pre-existing sites or is the significant increase only at novel sites?

The accession number was already present at the beginning of the Supplementary information file. (I am not sure about the rest) The antibody was used for ChIP seq before (<https://www.labome.com/product/Bethyl/A300-055A.html>). We validated the antibody by doing ChIP qPCR using reported cohesins binding sites.

8. Why is Supplementary Figure 4A, examining the levels of RAD21 and SMC1, included after numerous experiments examining the effects of RAD21 and CTCF knockdown. These should be included earlier in the manuscript.

"To investigate how c-Myc could increase cohesin chromatin occupancy, we tested if its activation affects the expression levels of cohesin subunits and regulators. Smc1, Smc3 and Rad21 protein levels did not change significantly upon c-Myc activation." We decided to put this piece of data here because we wanted to point out that the levels of the cohesin subunits do not change but the levels of the loaders do, in particular Mau2. For this reason, we investigated the effects of Mau2 knock down in c-Myc cells.

9. The authors need to check their statistical analysis. A few figures (e.g. Fig 4e) averages and SEMs between treatments are almost identical however the stats suggest the difference is highly significant. Moreover, the

authors have failed to describe how the intensity of SMC1, SMC3, H2AX etc are quantified. How many nuclei are counted? How many replicates? Are sample blinded before counting? how have the authors controlled for differences in fluorescent intensity between experiments?

We re-checked all statistical analysis. Regarding Figure 4e, the difference between sicont (-) and siMau2 Myc is statistically significant with a p-value of 0.0006 calculated with the Mann-Whitney test. The mean of the two populations is 56.55 ± 0.96 and 52.39 ± 1.1 (where the reported errors are the standard deviations of the mean). While the two distributions appear similar to the naked eye, their parameters are significantly different due to the large size of the samples (number of cells between 200 and 300). In Materials and Methods, we have added how many cells were analysed per sample. The number of replicates are indicated in each figure legend.

10. Reference 32 (main references) is incorrect and requires updating.

We have now corrected this reference.

Reviewers' comments:

Reviewer #1 (Remarks to the Author):

In this revised version, Peripolli et al., clarify some of the issues that I raised and provide some inferential argument in favor of their model. Yet, they do not clarify the main point of the paper, which is linking the increased genome occupancy by cohesins with the raise of replication stress. While They provide genetic evidence that higher or lower cohesion binding is associated with RS, the question of how the increase in genome bound cohesins generates RS is still unanswered. Is it due to a different distribution of the cohesin complex? Or is it that simply more cohesins bound (without differences in GW distribution) account for RS?

In their rebuttal to my review (see point 7), they write " Only a subset of these cohesins, together with CTCF, are more stably interacting with the DNA to establish the three-dimensional structure of chromatin, which happens in G1. Our data suggest that it is this particular subset of cohesins/CTCF that increases in a c-Myc-dependent manner, which is at the likely basis of causing RS in the subsequent S phase."

I do not see data in the manuscript that support or suggest such a claim, given the Authors have no indication of where in the genome the "extra" cohesins are loaded.

ChIP-seq analysis of cohesins would have provided useful insight, yet, rather surprisingly, the Authors state that "We agree with the reviewer that ChIP-seq data is not a robust quantitative assessment of chromatin occupancy.", thus dismissing the value of ChIP-seq as a way of mapping and evaluating genome-wide binding of cohesins. To my knowledge, ChIP-seq has been instrumental in clarifying the role of cohesins in genome-organization.

Without this information, the only claim supported by that manuscript is that "Myc is increasing replicative stress by increasing the amount of cohesion bound to chromatin".

By the way, I think the Authors misunderstood my comment 5 , where I stated: "ChIP-seq analyses need to be improved to be fully convincing: a snapshot of a genomic region and a general increase in peaks detection is not sufficient. As such, the data is not strong enough to support the claim that Myc activation promotes cohesins loading ". What I meant was to encourage the Authors to be more rigorous in the analysis of the ChIP-seq, which is quantitative if adequately performed and duly analysed.

Reviewer #3 (Remarks to the Author):

The revised manuscript by Peripolli et al., provides some additional data and discussion in line with the reviewer's comments. Unfortunately, these additions have only in part, clarified and substantiated some of the authors conclusions. Some of the new experiments and interpretation are difficult to understand without a concerted effort, making the reader work hard to interpret the data and form their own conclusions. Consequently, I reiterate my proposition, this study is too large for a short (2000 word) communication, even more so now. The novelty and intriguing link between MYC-activation, increased chromatin occupancy and replication stress requires substantially more discussion from the authors. Moreover, additional experiments are still required to validate the authors conclusions and provide mechanistic insight.

Major concerns:

1. The use of the CDK4/6 inhibitor to examine the effects of Myc-activation in G1 or S-phase on replication stress is a good addition. However, the authors have not gone far enough with the experiments and the figures are inappropriately labelled.

a. At first glance the labelling of Supplementary Figures 2G-J is very confusing, and the figure legend

only provides some experimental details. For example: What's does 4.5h or 7.5h mean in Supplementary Figures 2I-J? How do these correspond to Supplementary Figures 2G? Perhaps 4h and 7h or G1 and S-phase would have made more sense? The methods are included as supplementary information, with no word limit! Why haven't the authors included a comprehensive breakdown of Palbociclib and OHT treatment combinations?

b. If I understand Supplementary Figures 2J correctly, only MYC activation in G1 causes replication stress. This substantiates the authors assertion 'that c-Myc activity during G1 is required to cause RS in S phase.' The authors should quantify replication initiation events/origin firing in order to support the claim '.....data suggests that c-Myc-induced increase in origin firing does not cause RS per se.' The HU treatments, as acknowledge by the authors, is 'not ideal when studying oncogene-induced RS.', so conclusions based on this data are inappropriate.

2. I have major concerns about the interpretation of the CTCF and ChIP-seq experiments. As suggested by reviewer 1 and I, the ChIP-seq experiment still require further analysis to be fully convincing and relevant. What are the enrichment levels of SMC1 at CTCF sites (profile plots and/or heat maps)? Is the enrichment of SMC1 at the MYC-dependent CTCF sites similar to the pre-existing (common) loci? Are there any difference in the enrichment at the common sites? The authors have totally dismissed the loss of SMC1 at JunB, Fosl2 and Aft3 sites. Could the loss of cohesin binding cause replication stress because of altered chromatin structure or transcription?

3. The title of the manuscript ('Oncogenic c-Myc induces replication stress by increasing cohesins chromatin occupancy') suggests simply increasing cohesins chromatin occupancy results in replication stress. However, CTCF knockdown following MYC activation, does not prevent increased SMC1 loading on to chromatin (Figure 3F) whilst replication stress is 'completely rescued.' (Figure 3G-H). Actually, Figure 3F shows CTCF knockdown and Myc-activation causes greater SMC1 loading on to chromatin than MYC alone. Therefore, the title must be altered or more data added: these findings also need discussing further. CTCF is dispensable for cohesin loading onto DNA but is needed to enrich cohesin at specific binding sites. So, Cohesin (MYC-dependent) induced-replication stress may be determined by cohesin loading at specific loci. Although, the CTCF knockdown and SMC1 ChIP-seq experiments go some way to verifying this hypothesis, the evidence is circumstantial. Changes in chromatin structure and transcription following CTCF knockdown may alter the replication stress response upon MYC-activation. The manuscript requires some experiments examining the global effects of CTCF knockdown on SMC1 loading: ChIP-seq of SMC1 after CTCF knockdown. Presumably SMC1 is loaded at different sites, allowing the authors to speculate further on MYC-induced replication stress. For example altered proximity to Chromosomal fragile sites? Alternatively, the examination of gH2AX-ChIP-seq would demonstrate if 'excessive cohesin accumulation in an oncogenic context can interfere with the progression of the replisome' or the examination of SMC1 enrichment (ChIP-seq) following the knockdown of MAU2 would demonstrate any link between cohesin loading at CTCF sites and DNA replication stress.

4. In the abstract the authors write: 'Restoring the amount of chromatin-bound cohesins to control levels, or preventing the accumulation of cohesins at CTCF sites, in cells experiencing oncogenic c-Myc activity prevents replication stress.' The authors have provided no evidence to support the claim that.... 'preventing the accumulation of cohesins at CTCF sites.' Similarly, Figure 3 legend title requires modification.

5. In supplementary Figure 4, the authors have characterised the effects of over expressing Mau2 on replication stress. This is a crucial experiment for this study. Unfortunately, the western blotting data is unconvincing, which is a shame. Especially the high level of gH2AX in GFP-only cells (S4i). Here the addition of fibre data would be compelling, but I appreciate this is difficult in transiently transfected cells. Here the authors must demonstrate Mau-GFP is functional to corroborate their narrative: Mau2 increases cohesin loading on chromatin and this contributes to MYC-induced replication stress. Thus, immunofluorescence (or WB) quantification of SMC1/3 is required following Mau-GFP overexpression and Mau-GFP overexpression plus Rad21 knockdown.

6. In Figure 4H-K (and Suppl. Fig 4N-O) the authors have attempted to mitigate some of the reviewers concerns with additional data. These experiments are incomplete. Only two cancer cell lines are used (A549 and H1299), which have completely different karyotypes and genetic mutations/backgrounds, which could influence their response to siRad21 irrespective of MYC levels.

More cell lines must be examined to make these experiments relevant and convincing. Moreover, the proper supporting experiments are not included. In line with the authors hypothesis, chromatin bound SMC1/3 levels must be examined in these and other cell lines. Presumably H1299 has higher levels than A549?? Any differences in Mau2 levels?

Minor comments:

1. Please check figure legends, as some are incomplete.
2. Page 3, line 79: Numbers at the beginning of sentences should be spelt or the sentence re-phrased.
3. Labelling of Supplementary Figure 2G-J is confusing.
4. Numerical values on axis of all cell cycle profiles are illegible.

We would like to thank the reviewers for their useful comments aimed at increasing the impact of our study. We would also like to thank the editor for allowing us to address the reviewers concerns after revision. Specifically, the quantitative ChIP seq data, which is by no means standard practice, was extremely hard to obtain after the pandemic due to lack of funds, access to facilities and most importantly finding collaborators that were not just willing but more also able to help us with this. Whilst this has caused an extreme delay in completing the additional work the new data has significantly increased the impact of our study by establishing that c-Myc drives cohesins accumulation at all pre-existing sites, which, as previously shown, is predominantly CTCF dependent.

Our initial conclusion was that 'oncogenic c-Myc induces replication stress by increasing cohesins chromatin occupancy'. Based on the additional data, requested by the reviewers for the revision, we can now conclude that c-Myc affects increased accumulation of cohesins at CTCF sites and provide an indication that the c-Myc-dependent increase in Mau2 levels likely increases cohesin occupancy. This has produced a much more complete set of data that has allowed us to change the previously more general conclusion to 'Oncogenic c-Myc induces replication stress by increasing cohesins chromatin occupancy at CTCF sites'. Having address the concerns raised by the reviewers after revision we are hopeful our story can finally be accepted for publication.

Please find below our response to the reviewers.

Reviewer #1 (Remarks to the Author):

In this revised version, Peripolli et al., clarify some of the issues that I raised and provide some inferential argument in favor of their model. Yet, they do not clarify the main point of the paper, which is linking the increased genome occupancy by cohesins with the raise of replication stress. While They provide genetic evidence that higher or lower cohesion binding is associated with RS, the question of how the increase in genome bound cohesins generates RS is still unanswered. Is it due to a different distribution of the cohesin complex? Or is it that simply more cohesins bound (without differences in GW distribution) account for RS?

In their rebuttal to my review (see point 7), they write " *Only a subset of these cohesins, together with CTCF, are more stably interacting with the DNA to establish the three-dimensional structure of chromatin, which happens in G1. Our data suggest that it is this particular subset of cohesins/CTCF that increases in a c-Myc-dependent manner, which is at the likely basis of causing RS in the subsequent S phase.*". I do not see data in the manuscript that support or suggest such a claim, given the Authors have no indication of where in the genome the "extra" cohesins are loaded.

Without this information, the only claim supported by that manuscript is that "Myc is increasing replicative stress by increasing the amount of cohesion bound to chromatin".

We now include quantitative ChIP seq data pulling down the cohesion component SMC1 in siCTCF and siControl cells, with and without c-Myc (illustrated in Figure 2e and f, by relative binding between c-Myc and control profile plots and heat maps). These data indicate that the c-Myc-dependent increase in genome bound cohesion hyper-accumulates at CTCF sites and that this increase is CTCF dependent. This, together with our data that shows that loss of CTCF (siCTCF) prevents RS without decreasing the c-Myc-dependent increase in genome bound cohesion, allows us to conclude that c-Myc-dependent accumulation of cohesins on chromatin is not sufficient to cause RS, but also requires CTCF to facilitate the accumulation of cohesins at CTCF sites. We have now changed the title to reflect this new insight.

By the way, I think the Authors misunderstood my comment 5 , where I stated: "ChIP-seq analyses need to be improved to be fully convincing: a snapshot of a genomic region and a general increase in peaks detection is not sufficient. As such, the data is not strong enough to support the claim that Myc activation promotes cohesins loading ". What I meant was to encourage the Authors to be more rigorous in the analysis of the ChIP-seq, which is quantitative if adequately performed and duly analysed.

ChIPseq is a powerful technique to establish qualitative binding of proteins on chromatin via peak analysis (i.e. where proteins reside), but not very good at establishing absolute levels of protein bound (i.e. how much is bound to chromatin overall or at specific sites between different

conditions). To assess quantitatively how much is bound to chromatin requires normalization of the reads via an exogenous DNA spike-in to allow quantitative ChIP seq, which is what we now include. We agree with both reviewers that quantitative ChIP seq data was required to conclude that the c-Myc-dependent excess of cohesins accumulate at CTCF sites and that this is CTCF dependent. These data are now included and allows us to state this in the title, as suggested by reviewer 3.

Reviewer #3 (Remarks to the Author):

The revised manuscript by Peripolli et al., provides some additional data and discussion in line with the reviewer's comments. Unfortunately, these additions have only in part, clarified and substantiated some of the authors conclusions. Some of the new experiments and interpretation are difficult to understand without a concerted effort, making the reader work hard to interpret the data and form their own conclusions. Consequently, I reiterate my proposition, this study is too large for a short (2000 word) communication, even more so now. The novelty and intriguing link between MYC-activation, increased chromatin occupancy and replication stress requires substantially more discussion from the authors. Moreover, additional experiments are still required to validate the authors conclusions and provide mechanistic insight.

We would like to thank the reviewer for acknowledging the novelty and intriguing link between MYC-activation, increased chromatin occupancy [of cohesins] and replication stress established by our study. We agree that with the additional results our study has become extremely data heavy, but this is mostly because of additional supplementary data requested by the reviewers over time. However the key data, presented in the main figures, now allow us to draw much more specific and clear conclusions, which has made the paper overall easier to interpret and discuss.

Major concerns:

1. The use of the CDK4/6 inhibitor to examine the effects of Myc-activation in G1 or S-phase on replication stress is a good addition. However, the authors have not gone far enough with the experiments and the figures are inappropriately labelled.
 - a. At first glance the labelling of Supplementary Figures 2G-J is very confusing, and the figure legend only provides some experimental details. For example: What's does 4.5h or 7.5h mean in Supplementary Figures 2I-J? How do these correspond to Supplementary Figures 2G? Perhaps 4h and 7h or G1 and S-phase would have made more sense? The methods are included as supplementary information, with no word limit! Why haven't the authors included a comprehensive breakdown of Palbociclib and OHT treatment combinations?
 - b. If I understand Supplementary Figures 2J correctly, only MYC activation in G1 causes replication stress. This substantiates the authors assertion 'that c-Myc activity during G1 is required to cause RS in S phase.' The authors should quantify replication initiation events/origin firing in order to support the claim '.....data suggests that c-Myc-induced increase in origin firing does not cause RS per se.' The HU treatments, as acknowledge by the authors, is 'not ideal when studying oncogene-induced RS.', so conclusions based on this data are inappropriate.

We have now made these changes to the text to better reflect the data and make it easier to understand. Specifically, we have made changes to the labelling and to the figure legends and methods to make the data easier to understand and interpret.

2. I have major concerns about the interpretation of the CTCF and ChIP-seq experiments. As suggested by reviewer 1 and I, the ChIP-seq experiment still require further analysis to be fully convincing and relevant. What are the enrichment levels of SMC1 at CTCF sites (profile plots and/or heat maps)? Is the enrichment of SMC1 at the MYC-dependent CTCF sites similar to the pre-existing (common) loci? Are there any difference in the enrichment at the common sites? The authors have totally dismissed the loss of SMC1 at JunB, Fosl2 and Aft3 sites.

Based on our previous ChIPseq dataset we could not confidently conclude the enrichment levels of SMC1 at CTCF sites. Whilst our quantitative PCR ChIP did validate our conclusion, we now assess SMC1 binding via ChIP seq adding an exogenous DNA spike-in to allow quantitative ChIP seq, with and without CTCF. This indicates that c-Myc causes an enrichment of SMC1 at CTCF sites in an CTCF-dependent manner at the common sites, but also the c-Myc specific sites, which are also detected in non c-Myc conditions but were below our threshold detection in our previous non-

quantitative ChIP analysis. Additionally, the control specific peaks detected in our non-quantitative ChIP analysis, showing enrichment of JunB, Fos12 and Aft3 sites, are also detected in our quantitative ChIPseq analysis in c-Myc conditions. Interestingly also these peaks are CTCF-dependent. Overall, this new data indicates that c-Myc drives cohesins accumulation at all pre-existing sites, which, as previously shown, is predominantly CTCF dependent.

Could the loss of cohesin binding cause replication stress because of altered chromatin structure or transcription?

Our study shows that loss of the c-Myc-dependent excess of cohesion binding prevents RS. Other studies have shown that loss of chromatin bound cohesins does not *per se* causes RS but compromises a cells' ability to deal with RS and RS-induced DNA damage. It is thought that this is likely through a role in maintaining chromatin structure at stalled forks and/or double strand breaks.

3. The title of the manuscript ('Oncogenic c-Myc induces replication stress by increasing cohesins chromatin occupancy') suggests simply increasing cohesins chromatin occupancy results in replication stress. However, CTCF knockdown following MYC activation, does not prevent increased SMC1 loading on to chromatin (Figure 3F) whilst replication stress is 'completely rescued.' (Figure 3G-H). Actually, Figure 3F shows CTCF knockdown and Myc-activation causes greater SMC1 loading on to chromatin than MYC alone. Therefore, the title must be altered or more data added: these findings also need discussing further. CTCF is dispensable for cohesin loading onto DNA but is needed to enrich cohesin at specific binding sites. So, Cohesin (MYC-dependent) induced-replication stress may be determined by cohesin loading at specific loci. Although, the CTCF knockdown and SMC1 ChIP-seq experiments go some way to verifying this hypothesis, the evidence is circumstantial. Changes in chromatin structure and transcription following CTCF knockdown may alter the replication stress response upon MYC-activation. The manuscript requires some experiments examining the global effects of CTCF knockdown on SMC1 loading: ChIP-seq of SMC1 after CTCF knockdown.

We agree with the reviewer and with the addition of quantitative ChIPseq data we can now confidently concludes that it is c-Myc-dependent accumulation of cohesin at CTCF sites, in a CTCF-dependent manner, that is at the likely basis of RS. This, together with our data that shows that removing CTCF rescues c-Myc induced RS despite leaving high levels of cohesins on DNA, indicates that location of additional cohesins at CTCF sites is relevant for causing RS. As suggested by the reviewer we have now changed the title to reflect this new insight.

Presumably SMC1 is loaded at different sites, allowing the authors to speculate further on MYC-induced replication stress. For example altered proximity to Chromosomal fragile sites? Alternatively, the examination of gH2AX-ChIP-seq would demonstrate if 'excessive cohesin accumulation in an oncogenic context can interfere with the progression of the replisome' or the examination of SMC1 enrichment (ChIP-seq) following the knockdown of MAU2 would demonstrate any link between cohesin loading at CTCF sites and DNA replication stress.

Based on other published work from yeast and mammalian systems, it is likely the increased obstruction of additional cohesins at CTCF sites that interferes with the progression of the replisome causing replication forks to stall or slow down, which we discuss in the paper. Indeed, our work opens up many more questions that warrant further study, beyond the scope of the current work, which will be addressed by us and others to provide additional fundamental biological insight into cancer biology and it's treatment.

4. In the abstract the authors write: 'Restoring the amount of chromatin-bound cohesins to control levels, or preventing the accumulation of cohesins at CTCF sites, in cells experiencing oncogenic c-Myc activity prevents replication stress.' The authors have provided no evidence to support the claim that.... 'preventing the accumulation of cohesins at CTCF sites.' Similarly, Figure 3 legend title requires modification.

We have now addressed this with the additional quantitative ChIPseq data and made changes to the title to better reflect this new insight.

5. In supplementary Figure 4, the authors have characterised the effects of over expressing Mau2 on replication stress. This is a crucial experiment for this study. Unfortunately, the western blotting data is unconvincing, which is a shame. Especially the high level of gH2AX in GFP-only cells (S4i). Here the addition of fibre data would be compelling, but I appreciate this is difficult in transiently transfected cells. Here the authors must demonstrate Mau-GFP is functional to corroborate their narrative: Mau2 increases cohesin loading on chromatin and this contributes to MYC-induced replication stress. Thus, immunofluorescence (or WB) quantification of SMC1/3 is required following Mau-GFP overexpression and Mau-GFP overexpression plus Rad21 knockdown.

The focus of our work, and indicated by the title, is that c-Myc induces RS by increasing the amount of chromatin bound cohesins at CTCF sites. Upon the reviewers' request we carried out the OE Mau2 experiment, to further establish a potential mechanism by which c-Myc could increase cohesins chromatin occupancy. Since our Mau2 data is not exhaustive we have not included this in our title as a main conclusion, but presented as new insights that needs further investigation. We have now clearly indicated this in the text.

6. In Figure 4H-K (and Suppl. Fig 4N-O) the authors have attempted to mitigate some of the reviewers concerns with additional data. These experiments are incomplete. Only two cancer cell lines are used (A549 and H1299), which have completely different karyotypes and genetic mutations/backgrounds, which could influence their response to siRad21 irrespective of MYC levels. More cell lines must be examined to make these experiments relevant and convincing. Moreover, the proper supporting experiments are not included. In line with the authors hypothesis, chromatin bound SMC1/3 levels must be examined in these and other cell lines. Presumably H1299 has higher levels than A549?? Any differences in Mau2 levels?

The aim of our work was to establish mechanism(s) involved in c-Myc-induced RS. For this reason, we used a homogeneous system (RPE c-Myc), rather than a panel of cancer cell lines, allowing a more precise assessment of potential mechanisms. This revealed a novel mechanism for oncogene-induced RS. Upon the reviewers' request we extended this fundamental work to cancer cell lines, to provide an initial indication of a potential role for this mechanism in cancer. Since we agree that this data is not exhaustive, we have not include this in our title as a main conclusion, but have presented it as new insights that needs further investigation, which is clearly beyond the scope of the current work. We have now clearly indicated this in the text.

Minor comments:

1. Please check figure legends, as some are incomplete.
2. Page 3, line 79: Numbers at the beginning of sentences should be spelt or the sentence re-phrased.
3. Labelling of Supplementary Figure 2G-J is confusing.
4. Numerical values on axis of all cell cycle profiles are illegible.

We have now corrected these.

REVIEWER COMMENTS

Reviewer #1 (Remarks to the Author):

I will start by thanking the Authors for taking the time to address my previous remarks. As I understand, this entailed a substantial effort on their side.

This new version of the manuscript has been implemented by adding quantitative ChIP-seq data and new genetic evidence, which led to a revision of the model previously presented by the Authors. Based on these new data, the Authors propose that elevated Myc leads to the upregulation of Mau2, thus favoring cohesins loading and promoting CTCF loading. Contrary to what the Authors previously suggested, this increased association of cohesins to chromatin happens at all cohesins bound sites. Excessive CTCF loading on chromatin, but not excessive cohesins binding, supports Myc-induced RS (as the Authors argue in lines 238-240). The Authors also provide evidence that altered cohesins loading (by Mau2 over-expression) can induce RS.

Oddly, in the discussion, the Authors infer RS is due to excessive cohesins loading (line 324-325), yet this hypothesis seems to be confuted by the evidence that KD of CTCF bypasses Myc-induced RS without altering cohesins on chromatin (see also the Authors conclusion at line 238-240).

Overall, the manuscript provides evidence that excessive CTCF-loading is one of the factors leading to Myc-induced RS. Yet, fundamental questions remain unanswered as to how an increase of chromatin-bound CTCF would lead to RS, also in light of the consideration that CTCF knock-down experiments imply a cohesins-independent function of CTCF.

Thus, despite the new evidence and the interesting preliminary observations reported in this manuscript, overall, there has not been a significant advancement from the previous version.

Minor observations:

1. In their rebuttal, the Authors write that "Myc-dependent accumulation of cohesins on chromatin is not sufficient to cause RS, but also requires CTCF to facilitate the accumulation of cohesins at CTCF sites". I am afraid I did not see any evidence, in this manuscript, suggesting that CTCF facilitates the accumulation of cohesins.

2. Lines 27-30: I would suggest rephrasing more explicitly.

3. Fig2e,f: I would suggest revising the way SMC1 peaks were categorized, in particular the category (-). These are the peaks "detected" only in non-induced cells, yet (i) from the profile shown in (f), it is clear that the peak signals in "Myc" are as high as "UNT" and (ii) somewhat counter-intuitively, the violin plot in panel (e) shows that on these sites SMC1 binding is higher when Myc is induced (which is the opposite of what expected since according to the Authors these sites should be the one with more SMC1 binding in UNT conditions). As a side note, given the signal distribution, it would be more meaningful to report the median rather than the mean in fig2e.

Reviewer #3 (Remarks to the Author):

The re-submission by Cosetta Bertoli and Robertus A.M. de Bruin is an improved version of their manuscript however again I do not feel they have gone far enough to substantiate some of their main conclusions. In its current form this manuscript provides no mechanistic insight as to WHY increased cohesin loading on chromatin impairs DNA replication fork progression. On the contrary, the authors do persuasively demonstrate MYC overexpression induces increase SMC loading at CTCF sites. This is an interesting observation, providing another mechanism for replication stress. However, simply showing increased cohesin causes replication stress is somewhat descriptive. Therefore, I believe this study requires more mechanistic investigation in the form of additional experimentation.

Major

- The authors have not shown increase cohesions occupancy 'specifically' at CTCF sites induces replication stress. I suggest modifying their language in line with the data shown. For example, the title could read: Oncogenic c-Myc induces replication stress by increasing chromatin occupancy of cohesins in a CTCF manner. In the abstract the authors suggest 'excessive cohesins on chromatin at CTCF sites, which can interfere with the progression of replication forks.' The authors haven't provided any evidence for the slowing of forks or DNA damage at 'specific loci'. As I highlighted previously modulating levels or distribution of SMC1/3 or CTCF can influence genomic organisation or maintenance. Could an increase in replication stress and DNA damage be a consequence of unresolved DNA Topological Stress (PMID: 32259483)? Or is there direct conflict between SMC1/3 loading and replication forks. Neither of these have been investigated. I suggest the author modify some of their conclusions and include additional experiments. If they believe increased cohesins interfere with DNA fork progression why haven't they examined cohesins-fork collisions using PLA?
- On page 4 (Supplementary 1h), how was the cell cycle profiles determined? If this was performed using PI-Flow cytometry the authors must repeat this experiment using EdU/BrdU and PI (or similar). PI-Flow cytometry is incredibly inaccurate at distinguishing cell cycle phases. Indeed, in Supplementary fig. 1j, MYC activation increases p21 levels, which is commonly associated with cell cycle arrest, yet no difference is seen! Moreover, how do the authors account for the reduced colony formation? Presumably this is due to apoptosis since no change in proliferation is demonstrated. These experiments require further examination!
- In Supplementary 2C the percentage of origin firing are included in a graph on fork speed frequency. This is inadequate! The authors must show this data in a more appropriate manner: bar chart +/-SEM or SD from at least 3 replicates and with some statistics.
- The data provided in Supplementary Fig 2c-e require more explanation to concluded 'c-Myc-induced increase in origin firing does not cause RS per se.'
- On page 6 the authors write: 'c-Myc was activated, via addition of 4OH-T, either immediately after release (18 and 21 hours), therefore throughout G1 phase, or immediately before entering S phase (4.5 and 7.5 hours), and DNA damage and DNA fibres length were analyzed as above (Supplementary figure 2g-j).' Here there is no description of the data and reads more like methodology. These figures also require better labelling: release from what?
- For Supplementary fig. 2l, how many cells were counted? How many replicates did the authors perform? What was the concentration of DRB used? I am somewhat surprised DRB has no effect on MYC expressing cells!
- On page 7 the authors write: 'Overall, the distribution of cohesin in genic and intergenic regions does not change upon c-Myc activation (Fig. 2e).' Fig 2e does not show this!
- In Supplementary fig.3 m, n, the statistical comparisons are confusing. In my opinion MYC activation clearly increases DNA damage signalling irrespective of siRNA knockdown, yet the authors write 'RS-induced DNA damage was reduced (for Rad21 and CTCF depletion).' I do not agree! This data requires further explanation.

Minor

- The nomenclature of proteins and genes are incorrect throughout the manuscript.
- There are numerous instances of unnecessary preamble or overly complicated sentences that could either be eliminated or combined.
- o on page 5: 'To measure RS, we analyzed the length of DNA fibres.' Why tell the reader what you are going to do rather than getting to the point, especially in a short communication.
- Typos on page 5: 'indicative of slowing down or replication forks.'
- Change ChIPseq to ChIP-seq
- The new ChIP-seq data included in Fig 2 is too small.
- The data shows in (Fig. 2 e, f) fail to show that relative SMC1 binding (c-Myc/control) at ALL sites is higher in c-Myc cells compared to control cells and that this is CTCF-dependent (Fig. 2 e, f).
- I personally feel Fig 3e-h and supplementary figure 3K-l are the wrong way round: the surely the fibre data is more important for the narrative and would be more appropriate in the main figures!

We would like to thank the reviewers for their feedback, and the editor for their understanding and for allowing us to address the reviewers' concerns related to the current study. We have we added additional data on cell cycle state as suggested by reviewer #3 and made changes to the text to better reflect the data and make it easier to understand.

Reviewer #1 (Remarks to the Author):

I will start by thanking the Authors for taking the time to address my previous remarks. As I understand, this entailed a substantial effort on their side.

We sincerely thank the reviewer for appreciating our effort and taking the time to review the updated version of our manuscript.

This new version of the manuscript has been implemented by adding quantitative ChIPseq data and new genetic evidence, which led to a revision of the model previously presented by the Authors. Based on these new data, the Authors propose that elevated Myc leads to the upregulation of Mau2, thus favoring cohesins loading and promoting CTCF loading. Contrary to what the Authors previously suggested, this increased association of cohesins to chromatin happens at all cohesins bound sites. Excessive CTCF loading on chromatin, but not excessive cohesins binding, supports Myc-induced RS (as the Authors argue in lines 238-240). The Authors also provide evidence that altered cohesins loading (by Mau2 over-expression) can induce RS. Oddly, in the discussion, the Authors infer RS is due to excessive cohesins loading (line 324-325), yet this hypothesis seems to be confuted by the evidence that KD of CTCF bypasses Myc-induced RS without altering cohesins on chromatin (see also the Authors conclusion at line 238-240).

We have made changes to the conclusions to better reflect our data.

Overall, the manuscript provides evidence that excessive CTCF-loading is one of the factors leading to Myc-induced RS. Yet, fundamental questions remain unanswered as to how an increase of chromatin-bound CTCF would lead to RS, also in light of the consideration that CTCF knock-down experiments imply a cohesins-independent function of CTCF. Thus, despite the new evidence and the interesting preliminary observations reported in this manuscript, overall, there has not been a significant advancement from the previous version.

To clarify, our data supports a mechanism by which excessive cohesion loading (not CTCF-loading), and its subsequent hyperaccumulation at CTCF-dependent sites, leads to Myc-induced RS. Published work from yeast and mammalian systems suggests it is likely that increased obstruction caused by additional cohesins at CTCF sites interferes with the progression of the replisome leading to replication forks stalling or slowing-down, which we discuss in the paper. We agree that this needs further testing, but we hope the reviewer can understand that this is beyond the scope of the current work and will be the starting point for work by us and others to provide additional fundamental biological insight into cancer biology and its treatment.

Minor observations:

1. In their rebuttal, the Authors write that "Myc-dependent accumulation of cohesins on chromatin is not sufficient to cause RS, but also requires CTCF to facilitate the accumulation of cohesins at CTCF sites". I am afraid I did not see any evidence, in this manuscript, suggesting that CTCF facilitates the accumulation of cohesins.

We do not claim that CTCF facilitates the c-Myc-dependent increase in overall genome bound cohesin, but as mentioned in our previous rebuttal our quantitative ChIP-seq data does clearly show that CTCF facilitates the accumulation of cohesins at specific sites since siCTCF significantly reduces this accumulation.

2. Lines 27-30: I would suggest rephrasing more explicitly.

We have now changed this to "However, we found that the accumulation of cohesin on chromatin is not sufficient to cause replication stress, but also requires for the increased chromatin bound cohesin to accumulate at specific sites in a CTCF-dependent manner", which we hope the reviewer agrees is more explicit.

3. Fig2e,f: I would suggest revising the way SMC1 peaks were categorized, in particular the category (-). These are the peaks "detected" only in non-induced cells, yet (i) from the profile shown in (f), it is clear that the peak signals in "Myc" are as high as "UNT" and (ii) somewhat counter-intuitively, the violin plot in panel (e) shows that on these sites SMC1 binding is higher when Myc is induced (which is the opposite of what is expected since according to the Authors these sites should be the one with more SMC1 binding in UNT conditions). As a side note, given the signal distribution, it would be more meaningful to report the median rather than the mean in fig2e.

We discussed this very same issue at great length with the team and in the end agreed to keep the categories identified by our non-quantitative ChIP-seq for two reasons. First and foremost, it shows the importance of quantitative ChIP-seq versus the standard practice of non-quantitative ChIP-seq, as rightly pointed out by both reviewers. Our new quantitative ChIP-seq shows that relative SMC1 binding (c-Myc/control) is higher in c-Myc cells compared to control cells, at all sites detected in our non-quantitative ChIP-seq experiment, which was missed by non-quantitative ChIP-seq. However, keeping the categories from our non-quantitative ChIP-seq also shows that the (-) category is different (less binding and still CTCF-dependent without CTCF site enrichment), which would have been missed if all peaks were combined based on our quantitative ChIP-seq. Therefore, by keeping the categories identified by our non-quantitative ChIP-seq we both show the power of quantitative ChIP-seq for establishing absolute levels of protein bound (i.e. how much is bound to chromatin overall or at specific sites between different conditions), but also show the importance of establishing qualitative binding of proteins on chromatin via peak analysis (i.e. where proteins reside). We hope that the reviewer and editor agree with our decision.

Reviewer #3 (Remarks to the Author):

The re-submission by Cosetta Bertoli and Robertus A.M. de Bruin is an improved version of their manuscript however again I do not feel they have gone far enough to substantiate some of their main conclusions. In its current form this manuscript provides no mechanistic insight as to WHY increased cohesin loading on chromatin

impairs DNA replication fork progression. On the contrary, the authors do persuasively demonstrate MYC overexpression induces increase SMC loading at CTCF sites. This is an interesting observation, providing another mechanism for replication stress. However, simply showing increased cohesin causes replication stress is somewhat descriptive. Therefore, I believe this study requires more mechanistic investigation in the form of additional experimentation.

We thank the reviewer for acknowledging we “persuasively demonstrate MYC overexpression induces increase SMC loading at CTCF sites. This is an interesting observation, providing another mechanism for replication stress. “

Major

- The authors have not shown increase cohesins occupancy ‘specifically’ at CTCF sites induces replication stress. I suggest modifying their language in line with the data shown. For example, the title could read: Oncogenic c-Myc induces replication stress by increasing chromatin occupancy of cohesins in a CTCF manner. In the abstract the authors suggest ‘excessive cohesins on chromatin at CTCF sites, which can interfere with the progression of replication forks.’ The authors haven’t provided any evidence for the slowing of forks or DNA damage at ‘specific loci’. As I highlighted previously modulating levels or distribution of SMC1/3 or CTCF can influence genomic organisation or maintenance. Could an increase in replication stress and DNA damage be a consequence of unresolved DNA Topological Stress (PMID: 32259483)? Or is there direct conflict between SMC1/3 loading and replication forks. Neither of these have been investigated. I suggest the author modify some of their conclusions and include additional experiments. If they believe increased cohesins interfere with DNA fork progression why haven’t they examined cohesins-fork collisions using PLA?

We would like to thank the reviewer for their suggestion of modifying the title to better reflect our data. We have now changed the title to “Oncogenic c-Myc induces replication stress by increasing cohesins chromatin occupancy in a CTCF-dependent manner”. We do agree that we have not provided direct evidence of slowing of replication forks at specific sites and have changed the text to better reflect this. We agree that this needs further testing, but we hope the reviewer can understand that this is beyond the scope of the current work and will be the starting point for work by us and others to provide additional fundamental biological insight into cancer biology and its treatment.

- On page 4 (Supplementary 1h), how was the cell cycle profiles determined? If this was performed using PI-Flow cytometry the authors must repeat this experiment using EdU/BrdU and PI (or similar). PI-Flow cytometry is incredibly inaccurate at distinguishing cell cycle phases. Indeed, in Supplementary fig. 1j, MYC activation increases p21 levels, which is commonly associated with cell cycle arrest, yet no difference is seen! Moreover, how do the authors account for the reduced colony formation? Presumably this is due to apoptosis since no change in proliferation is demonstrated. These experiments require further examination!

We agree that PI-Flow cytometry is inaccurate and now include cell cycle experiments including EdU incorporation (Supplementary 1h), as suggested by the reviewer. The new data does indeed show the general trend better than indicated by the PI-Flow cytometry data. It

shows that the effect of c-Myc activity on the cell cycle is progressive. Activation of c-Myc causes an increase in S phase population (likely through shorting of G1) in the first 24h, which is in line with its pro-proliferation activity. However, at 48h c-Myc cells show increased arrest in G1, likely because of an increase in p21 levels lengthening the G1 phase, which is the consequence of the accumulation in RS-induced DNA damage that activates the DDR and consequently causes a cell cycle arrest. This is in line with our data on colony formation, measured many days after c-Myc activation, which shows a decrease in the number of colonies of cells experiencing c-Myc activity compared to control.

- In Supplementary 2C the percentage of origin firing are included in a graph on fork speed frequency. This is inadequate! The authors must show this data in a more appropriate manner: bar chart +/-SEM or SD from at least 3 replicates and with some statistics.

And

- The data provided in Supplementary Fig 2c-e require more explanation to concluded 'c-Myc-induced increase in origin firing does not cause RS per se.'

We agree that our data on origin firing is not strong enough for supporting the conclusion that deregulation of origin firing is not involved, therefore we removed this data and changed the text accordingly.

- On page 6 the authors write: 'c-Myc was activated, via addition of 4OH-T, either immediately after release (18 and 21 hours), therefore throughout G1 phase, or immediately before entering S phase (4.5 and 7.5 hours), and DNA damage and DNA fibres length were analyzed as above (Supplementary figure 2g-j).' Here there is no description of the data and reads more like methodology. These figures also require better labelling: release from what?

We changed the text to make it clearer.

- For Supplementary fig. 2l, how many cells were counted? How many replicates did the authors perform? What was the concentration of DRB used? I am somewhat surprised DRB has no effect on MYC expressing cells!

The experiment was repeated three time as stated in the figure legend. We have added the DRB concentration used in the methods and stated that at least 250 cells were analysed per condition in each repeat. DRB treatment does reduce EU levels in Myc cells as shown in the images and quantifications.

- On page 7 the authors write: 'Overall, the distribution of cohesin in genic and intergenic regions does not change upon c-Myc activation (Fig. 2e).' Fig 2e does not show this!

We have removed this.

- In Supplementary fig.3 m, n, the statistical comparisons are confusing. In my opinion MYC activation clearly increases DNA damage signalling irrespective of siRNA knockdown, yet the authors write ‘RS-induced DNA damage was reduced (for Rad21 and CTCF depletion).’ I do not agree! This data requires further explanation.

We changed the text to better reflect the data and grouped the control (-) and c-Myc (Myc) data to better indicate the statistical comparisons, which shows that CHK1-p and gamma-H2AX are significantly lower upon Rad21 and CTCF depletion.

Minor

- The nomenclature of proteins and genes are incorrect throughout the manuscript.

This has been corrected in the text and figures.

- There are numerous instances of unnecessary preamble or overly complicated sentences that could either be eliminated or combined.
o on page 5: ‘To measure RS, we analyzed the length of DNA fibres.’ Why tell the reader what you are going to do rather than getting to the point, especially in a short communication.

We prefer to tell the story, but if required we will address this if we must reduce the wordcount to fit the communication format.

- Typos on page 5: ‘indicative of slowing down or replication forks.’

We have corrected this typo.

- Change ChIPseq to ChIP-seq.

We have made this change.

- The new ChIP-seq data included in Fig 2 is too small.

We increased the size of this panel.

- The data shows in (Fig. 2 e, f) fail to show that relative SMC1 binding (c-Myc/control) at ALL sites is higher in c-Myc cells compared to control cells and that this is CTCF dependent.

The ratio between cMyc and untreated is above 1, indicating an increased binding in c-Myc-induced compared to control. This is reduced by CTCF silencing, supporting a dependence on CTCF.

(Fig. 2 e, f).

- I personally feel Fig 3e-h and supplementary figure 3K-l are the wrong way round: the surely the fibre data is more important for the narrative and would be more appropriate in the main figures!

The fibre data is included in the main (Figure 3g-h). In the supplementary we added the two extra repeats of the experiment.

REVIEWERS' COMMENTS

Reviewer #3 (Remarks to the Author):

Despite answering most of the concerns raised, this manuscript provides no mechanistic insight in to how an increase of chromatin-bound cohesin leads to RS. I disagree with the authors that additional mechanistic experiments are outside the scope of this study! Three years have passed since the first submission, which is ample time to explore how accumulated chromatin-bound cohesin leads to RS. For example, Proximity Ligation Assays (PLA) between SMC1/3 (cohesion) and PCNA (fork) would be a perfect addition to this study. Such experiments would be most insightful, especially since the authors STILL, without any evidence, suggest that the accumulation of cohesion interferes with the progression of the replisome or replication forks (see lines 31-33 and 316-316).

We acknowledge that reviewer #3 would like us to extend our study even further, to provide still additional evidence for our model. We would like to thank the editor for allowing us to address the reviewers' concerns related to the current study. Whilst we agree that more can be done, we strongly feel that this is beyond the scope of the current study and thank the editor for their understanding.

Reviewer #3 (Remarks to the Author):

Despite answering most of the concerns raised, this manuscript provides no mechanistic insight in to how an increase of chromatin-bound cohesin leads to RS. I disagree with the authors that additional mechanistic experiments are outside the scope of this study! Three years have passed since the first submission, which is ample time to explore how accumulated chromatin-bound cohesin leads to RS. For example, Proximity Ligation Assays (PLA) between SMC1/3 (cohesion) and PCNA (fork) would be a perfect addition to this study. Such experiments would be most insightful, especially since the authors STILL, without any evidence, suggest that the accumulation of cohesion interferes with the progression of the replisome or replication forks (see lines 31-33 and 316-316).

We do agree that we have not provided direct evidence of slowing of replication forks at specific sites and have changed the text to better reflect this. However, as acknowledge by all reviewers, our study establishes a new mechanism of oncogene-induced replication stress, showing that oncogenic c-Myc induces replication stress by increasing cohesins chromatin occupancy in a CTCF-dependent manner. Whilst our data suggests that the increased accumulation of cohesins at CTCF site is likely to interfere with the progression of replication forks, contributing to oncogene-induced replication stress we agree that this needs further testing. Sharing our work through publication in Nature Communications will be the starting point for work by us and others to provide additional fundamental biological insight into cancer biology and its treatment.